# *PoliCon*: Evaluating LLMs on Achieving Diverse Political Consensus Objectives

**Zhaowei Zhang**[1] ✉, **Xiaobo Wang**[2,5], **Minghua Yi**[3], **Mengmeng Wang**[5], **Fengshuo Bai**[4,6],
**Zilong Zheng**[5‡], **Yipeng Kang**[5‡] ✉, **Yaodong Yang**[1‡]

[1] Institute for Artificial Intelligence, Peking University    [2] USTC    [3] WHU    [4] SJTU
[5] State Key Laboratory of General Artificial Intelligence, BIGAI    [6] Zhongguancun Academy

## Abstract

Achieving political consensus is crucial yet challenging for the effective functioning of social governance. However, although frontier AI systems represented by large language models (LLMs) have developed rapidly in recent years, their capabilities in this scope are still understudied. In this paper, we introduce *PoliCon*, a novel benchmark constructed from 2,225 high-quality deliberation records of the European Parliament over 13 years, ranging from 2009 to 2022, to evaluate the ability of LLMs to draft consensus resolutions based on divergent party positions under varying collective decision-making contexts and political requirements. Specifically, *PoliCon* incorporates four factors to build each task environment for finding different political consensus: specific political issues, political goals, participating parties, and power structures based on seat distribution. We also developed an evaluation framework based on social choice theory for *PoliCon*, which simulates the real voting outcomes of different political parties to assess whether LLM-generated resolutions meet the requirements of the predetermined political consensus. Our experimental results demonstrate that even state-of-the-art models remain undersatisfied with complex tasks like passing resolutions by a two-thirds majority and addressing security issues, while uncovering their inherent partisan biases and revealing some behaviors LLMs show to achieve the consensus, such as prioritizing the stance of the dominant party instead of uniting smaller parties, which highlights *PoliCon*'s promise as an effective platform for studying LLMs' ability to promote political consensus. The code is released at ⌂ PoliCon.

## 1 Introduction

One of the fundamental prerequisites for effective social governance is establishing political consensus across diverse stakeholders (Prothro & Grigg, 1960; Lijphart et al., 1999; Huckfeldt et al., 2004; Rawls, 2020). From infrastructure development to welfare policies, consensus-building underpins the legitimacy (Cohen, 2005) and implementation of collective decisions (Citrin, 2001; Potapchuk & Crocker, 2017; Shehu, 2017). Yet, in pluralistic societies, conflicting values, power dynamics, and issue complexity render this process exceptionally challenging (Raiffa, 1982; Ehtamo et al., 1999; Susskind et al., 1999; Baker & Azher, 2024). While large language models (LLMs) have shown promise in facilitating group discussions (Chiang et al., 2024), supporting democratic deliberation (Small et al., 2023; Fish et al., 2023; Tessler et al., 2024; Jarrett et al., 2025), resolving regional conflicts (Konya et al., 2025), and analyzing ideological stances (Chen et al., 2024; Kim et al., 2025), their capacity to find consensus in real and complex political scenarios remains underexplored. This gap raises a critical question: ***Can LLMs bridge divides among divergent stakeholders and achieve the objectives of different political consensus in real-world settings?***

To study this problem, in this paper, we introduce *PoliCon*, a benchmark constructed based on 2,225 real deliberation records of the European Parliament over a 13-year period ranging from 2009 to 2022. It is worth noting that the purpose of *PoliCon* is not to simulate the European Parliament, but rather to leverage these data to transform complex real-world political deliberation processes into a systematic research framework for evaluating the ability of LLMs under different consensus objectives. To

---

‡ Corresponding Authors.

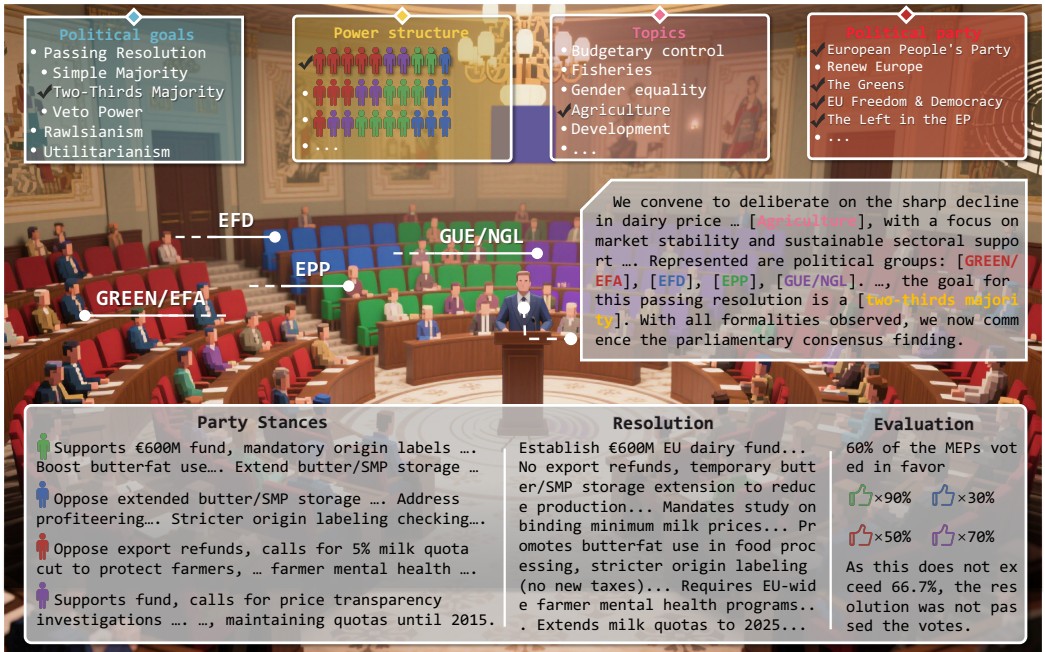

Figure 1: **An example scenario in *PoliCon*.** In each task, *PoliCon* builds a collective decision-making environment with varying political goals, power structures, issues, and participating parties. The tested LLM then attempts to achieve a consensus resolution based on these setups and the divergent party positions. The outcome is evaluated first via a simulated vote and then mapped to a quantitative score according to the specific environment setting by *PoliCon*'s evaluation framework.

enhance its generality, we further design diverse environment settings to ensure that it can flexibly adapt to various consensus objectives and real-world collective decision-making scenarios.

Specifically, *PoliCon* has designed four adjustable factors to build different task environments, which are: (1) **Political issues**: the political problems to be discussed and their topic classification, (2) **Political goals**: the criteria for achieving the corresponding political consensus, (3) **Participating parties**: different numbers of involving stakeholders with varying stances, and (4) **Power structures**: differences in influence and discourse power of each party due to their number of seats. By combining these settings, we have constructed a total of 28,620 detailed scenarios. To assess whether LLM-generated resolutions meet the corresponding political goals, we further developed an open-ended evaluation framework in *PoliCon* based on the social choice theory (Sen, 1986; Kelly, 2013). Through our experiments, we have verified its strong capability to simulate the real voting results, thereby allowing the effective evaluation in *PoliCon* (subsection 4.1).

We illustrate one of the *PoliCon*'s task scenarios in Figure 1. The upper part presents the setting of the current collective decision-making environment. The seating colors in the figure represent the seat distribution among the four participating parties, which are GREEN/EFA (red, 50%), EPP (green, 20%), GUE/NGL (purple, 20%), and EFD (blue, 10%). The lower part demonstrates the LLM's consensus-finding process. The parliamentary president announced the need to discuss the issue of surplus dairy products and introduced the political goal is to passing the resolution with a two-thirds majority among the members of the European Parliament (MEPs).

Subsequently, each participating party expresses inconsistent positions on this issue. For example, EPP and EFD have significant disagreements on the matter of extending storage time, while GREEN/EFA and GUE/NGL have differences over export refunds. Although the resolution generated by the evaluated LLM partially considered the apportionment of seats among different parties to balance conflicting positions, it still failed to reconcile a new consensus resolution beyond the compromise. As a result, in *PoliCon*'s evaluation, 50%, 90%, 70%, and 30% of the MEPs from GREEN/EFA, EPP, GUE/NGL, and EFD vote in favor of the resolution, respectively. Considering the seat distribution, only 60% of the entire parliament voted in favor of the resolution, which does not meet the two-thirds majority standard, and thus the resolution was not passed.

We perform a comprehensive evaluation using *PoliCon* in six representative LLMs, revealing notable variations in their ability to find political consensus (subsection 4.2). While most LLMs perform well on simple majority tasks, they struggle with more difficult challenges, such as passing resolutions with a two-thirds majority or addressing security issues (subsection 4.3). Furthermore, our analysis uncovers the inherent biases in the tested LLMs, which can help explain their behaviors and provide empirical guidance for deploying them in real-world scenarios (subsection 4.4).

In summary, this paper makes four major contributions. **Firstly**, we conduct a large-scale scraping and thorough cleaning of a vast amount of European Parliament deliberation records, compiling 2,225 high-quality complete parliamentary records. **Secondly**, we define the problem of evaluating LLMs' ability to find political consensus and construct *PoliCon* based on these records. **Thirdly**, we develop an open-ended evaluation framework that can simulate the proportion of MEPs who vote in favor in each party. **Lastly**, we demonstrate that *PoliCon* can well assess the LLMs' ability to find diverse political consensus, highlighting its promise as an effective research platform for the realm.

## 2 RELATED WORK

Achieving political consensus, due to its realistic and complex scenarios, the conflict of stances and values, and the need to consider diverse power structures, differs from existing works that primarily consider conversational grounding (Udagawa & Aizawa, 2019; 2020; 2021; Mitsuda et al., 2022; Mohapatra et al., 2024) and game-theoretic bargaining (Lewis et al., 2017; Paquette et al., 2019; Zhou et al., 2023; Bianchi et al., 2024; Huang et al., 2024; Xia et al., 2024; Abdelnabi et al., 2024), becoming a novel and challenging problem. To our knowledge, there are currently no studies that construct a benchmark to evaluate LLMs' ability to achieve different objectives of political consensus, but there are some works that have explored LLMs for democratic deliberation and benchmarks.

**LLMs for Democratic Deliberation.**  LLMs' powerful semantic processing capabilities (He et al., 2025; Li et al., 2025; Kang et al., 2020) and broad value alignment potential (Zhang et al., 2026; Ziheng et al., 2026; Kang et al., 2025) in social scenarios (Ziheng et al., 2025; Smith et al., 2025) have led some studies to explore how they can accelerate the process of democratic deliberation. Konya et al. (2023) design a pipeline allowing LLMs to participate in every stage of democratic elections, aiding in extracting and summarizing complex texts to improve decision-making efficiency. Fish et al. (2023) utilizes LLMs' generative abilities to synthesize a set of opinions most satisfactory to the majority based on survey results about chatbot personalization. Small et al. (2023) apply LLMs to the deliberation platform Polis (Small et al., 2021), finding that LLMs enhance efficiency but still pose unresolved risks. Bakker et al. (2022); Tessler et al. (2024) fine-tune LLMs to repeatedly generate and refine statements representing a group's collective stances on social or political issues.

**Benchmarks in Political Settings.**  LLMs have been widely applied to political science tasks (Li et al., 2024). However, political science covers a wide range of research questions, resulting in diverse benchmarks. Kornilova & Eidelman (2019); Arregui & Perarnaud (2022); KlÄijver et al. (2023); Shu et al. (2024) provide data on texts and the ideologies of their associated political parties, which are used for semantic analysis of texts covering different ideologies. Garzia et al. (2017); Vamvas & Sennrich (2020) extensively collect public comments on various political issues in Europe to study the positioning and classification of political positions. Kornilova & Eidelman (2019); Shu et al. (2024); Arregui & Perarnaud (2022) provide a large collection of legal text data, facilitating research in the generation and summarization of legal documents. POLCA (Moghimifar et al., 2024) collects party statements from several European countries to evaluate whether LLMs can determine if a statement is likely to appear in the final agreement. Stammbach et al. (2024); Chalkidis & Brandl (2024); Batzner et al. (2024) investigate whether LLMs have intrinsic political bias and explore the impact of fine-tuning and prompting on their political stance. Liang et al. (2025) constructs a benchmark based on the United Nations resolution process to evaluate whether LLMs can accurately capture the political stances of member states, simulate voting, and emulate delegate speeches. Although these works offer benchmarks for political science research, their focus is not on studying the ability of LLMs to achieve the objectives of political consensus.

## 3 *PoliCon* BENCHMARK

How to effectively evaluate LLMs' capability to achieve different objectives of political consensus is a significant challenge. To address this problem, we seek a benchmark that satisfies the following requirements: (1) **Authenticity**: All data must come from real political scenarios; (2) **Conflict**:

Under each political issue, there must be a varying number of parties holding different opinions; (3) **Diverse Power Structures**: The generated political consensus need to consider the impact of different parties; (4) **Various Political Goals**: The state when political consensus is achieved; (5) **Open-ended Evaluation**: It should be capable of automatically evaluating the quality of the political consensus generated by LLMs in scenarios that meet the above requirements.

In the following sections, we will provide a detailed introduction to *PoliCon* and how it addresses the above challenges. In subsection 3.1, we will describe the data collection and cleaning procedures of *PoliCon*, explaining why it meets the criterion of authenticity and open-ended evaluation. In subsection 3.2, we will explain why it addresses the remaining challenges by detailing the different task definitions and how we construct these tasks based on the collected data.

## 3.1 DATA COLLECTION PROCEDURE

We conduct a large-scale scraping and combine data sourced from the official website of the European Parliament[1] , HowTheyVote[2] , and the VoteWatch Europe dataset (HIX et al., 2022), to obtain a comprehensive collection of parliamentary records from the European Parliament spanning a 13-year period from 2009 to 2022. This dataset covers three parliamentary terms of the 7th, 8th, and 9th, as well as includes detailed information on issues, topics, debates, resolutions, and votes.

Unlike previous datasets that were also collected from the European Parliament or political parties (Koehn, 2005; HIX et al., 2022; Chalkidis & Brandl, 2024; Moghimifar et al., 2024), we (1) do not just scrape a single aspect of the parliamentary process, such as debates (Chalkidis & Brandl, 2024) or votes (HIX et al., 2022), but instead collect all information corresponding to each issue from different sources separately, further aligning and integrating them more comprehensively, including information on issues, topics, debates, resolutions, and votes. (2) We perform additional cleaning and post-processing on the data to enhance its quality and readability. (3) The cleaned voting and resolution data can serve as the basis for our open-ended evaluation, allowing further verification of whether our designed evaluation framework aligns with real-world voting outcomes. These contributions not only enhance the quality and diversity of our data but also allow the data to transcend the scope of a single task (such as being used solely for text translation (Koehn, 2005)) and further enable the construction of various complex political tasks and scenarios in *PoliCon*. We will introduce them one by one as follows:

**Data Collection.** We first match the URL provided for each issue's voting information in the VoteWatch Europe dataset with the corresponding issue URLs on the European Parliament's official website and HowTheyVote. This allows us to obtain the issue and resolution content corresponding to each voting record. We further match the resolution with the debate URL on the European Parliament's website using the issue name, enabling us to scrape the corresponding debate information. In this way, we obtain 30,698 raw parliamentary records. However, since many records were incomplete or duplicated, we further refine the data, retaining only those where the final vote was confirmed to be finished and all information was complete. The detailed filtering steps are provided in Appendix A.1. Furthermore, referring to the topics defined in the VoteWatch Europe dataset (HIX et al.,

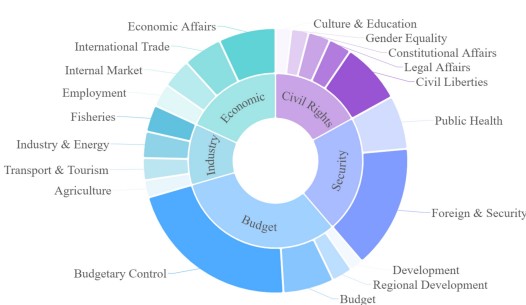

Figure 2: The 5 coarse-grained and 19 fine-grained topic categories of issues in *PoliCon*, whose definitions can be found in Appendix B.1. The shade of the color indicates the proportion of the fine-grained topic within the coarse-grained topic; the darker the color, the higher the proportion.

2022), we classify all these complete data records into 5 coarse- and 19 fine-grained topics (detailed in Figure 2), such as "culture & education", "agriculture", "international trade", etc. Through this approach, we integrate different pieces of information on the same issue from various sources, ultimately selecting 2,225 complete, high-quality raw data entries, ensuring that each data entry contains a quintuple of raw information: (issue, topic, debates, resolution, votes).

---

[1]https://www.europarl.europa.eu
[2]https://howtheyvote.eu

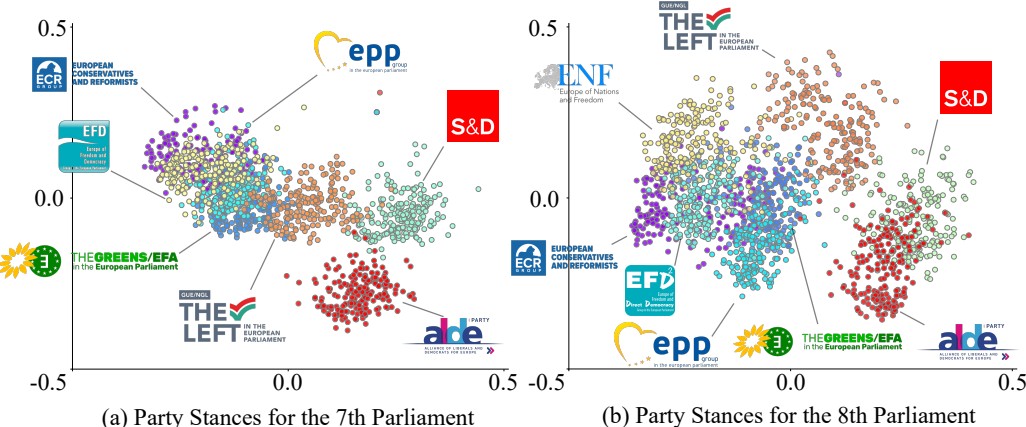

(a) Party Stances for the 7th Parliament          (b) Party Stances for the 8th Parliament

Figure 3: Semantic representation distribution of party stances (indicated by their symbols) in the 7th (2009-2014) and 8th (2014-2019) terms of the European Parliament in *PoliCon*.

**Data Cleaning.** To address raw data redundancy, we employ DeepSeek-R1 (Guo et al., 2025) and rule-based methods for data cleaning. DeepSeek-R1 is used to summarize the relevant background of the issue, select party stances from debates, and remove redundant content from the resolution. Rule-based methods are then applied to randomly perform synonym replacement to diversify the stance data. Voting data is processed by matching each member with their party and calculating party voting results by rounding down the proportion of MEPs within the party who voted in favor to an integer between 0 and 9. This results in cleaned sextuples of (issue, topic, background, stances, resolution, votes) containing relevant party information. Further details of the data cleaning procedure, prompts, and cleaning qualification can be found in Appendix A.6, Appendix C.1, and Appendix D.

**Constructing the Open-ended Evaluation.** Based on the sextuple data, we can perform the open-ended evaluation for each party's voting results on each issue by inputting the background of the issue, each party's stances, and the resolution generated by the evaluated LLM.

Our evaluation framework consists of two parts. The first part adopts the LLM-as-a-judge approach to conduct a simulated vote for each political party. The outcome of this simulation is represented as a scalar score between 0 and 9, indicating the proportion of the MEPs within the party voting in favor. We define the $n$ parties participating in each issue as $P = \{p_1, p_2, \ldots, p_n\}$. For each party $p_i$, its stance is represented as $s_i$. The corresponding voting score $u_i$ for the party can be calculated using $u_i = \text{JUDGE}(\cdot \mid \text{background}, s_i, \text{resolution})$, where $u_i \in \{0, 1, 2, 3, 4, 5, 6, 7, 8, 9\}$. This score takes into account both the resolution's alignment with the party's stance and its feasibility under the given background, including factors such as resource requirements (e.g., funds and energy) and internal conflicts. The specific evaluation prompt can be found in Appendix C.3. In subsection 4.1, we verified that its simulation is highly consistent with real-world parliamentary voting results.

The second part is the political consensus evaluation module, which maps all the votes into quantitative scores of different political consensus objectives based on social choice theory. In *PoliCon*, we define a variety of tasks to evaluate different political consensus objectives that may arise in real-world collective decision-making scenarios. Depending on the task definition, the computation of this module also varies. We provide detailed descriptions of these settings in subsection 3.2.

## 3.2 TASK SETTINGS

After collecting and cleaning the raw data, we further expand and organize these data to construct different task settings for each issue in *PoliCon*. These settings are designed to construct scenarios of conflict, diverse power structures, and various political goals, in order to meet the evaluation needs of different political consensus objectives that may arise in real-world collective decision-making tasks. In the following paragraphs, we will introduce each aspect separately:

**Participating Stakeholders.** The core requirement of conflict is to have a different number of stakeholders with various positions on each issue. There are clear differences in political positions among the parties in the European Parliament (McElroy & Benoit, 2007; Proksch & Slapin, 2010; McElroy & Benoit, 2012). To demonstrate this point more obviously, we randomly sample 200

stance data points from each party during the 7th and 8th parliamentary terms. We then use OpenAI's text-embedding-003-small (OpenAI, 2024) model to map each party's stances into a semantic representation space, and employ Principal Component Analysis (PCA) (Wold et al., 1987) to visualize this information. As shown in Figure 3, the stances of each party form distinct clusters in the semantic space, with significant differences (detailed in Appendix B.2). We further design three different settings for the number of participating parties in *PoliCon*: 2, 4, and 6. For each issue, we select the corresponding number of parties with the highest voting variance to enhance the conflict.

**Power Structure.** One major challenge of finding political consensus is dealing with complex power structures. To more accurately simulate this feature in reality, we allocate seats to each participating party in the current parliament scenario to demonstrate their political influence. We define the calculation of the total votes in favor MEP number $u$ in this setting as:

$$u = \sum_{i=1}^{n} w_i u_i, \tag{1}$$

where $w_i$ represents the proportion of seats occupied by party $p_i$ in the parliament, satisfying $\sum_{i=1}^{n} w_i = 1$ and $\forall w_i \geq 0$.

Since our goal is to explore whether LLMs can effectively assist in reaching political consensus under diverse scenarios, in constructing the tasks, we randomly assign each party's seats in the parliament. This approach not only enriches our task settings but also subtly incorporates potential biases of LLMs toward different parties into our evaluation framework. Specifically, if the tested LLM holds a bias towards certain parties, it may fail to reach the political consensus that meets the requirements when those parties become the dominant ones, and this will be reflected in the final results. In practical use of *PoliCon*, users can also adjust this setting according to their own needs.

**Voting Mechanism.** We refer to the collective decision-making procedure in different parliaments and the United Nations Security Council (UNSC) to set three common voting mechanisms, which are: (1) **Simple Majority**: A resolution needs to be voted through by more than 50% of the parliamentary seats. We define the boolean variable $v \in \{0, 1\}$ to indicate whether the resolution will be passed under this voting mechanism. In the setting of simple majority, $v = 1$ only if $u \geq 5$. (2) **Two-thirds Majority**: A resolution needs to be voted through by more than two-thirds of the parliamentary seats. In this setting, $v = 1$ only if $u \geq 6.67$. (3) **Veto Power**: To extend *PoliCon* beyond the context of the European Parliament, we introduce a veto mechanism. In the UNSC, permanent members have veto power (United Nations). In our setting, the tested LLM needs to generate a resolution that can be passed by a simple majority of MEPs in the parliament and not be rejected by the vetoing party (in favor rate over 60%). In this setting, $v = 1$ only if $u \geq 5$ and $u_k \geq 6$, where $u_k$ is the voting score of the vetoing party. In the actual process of constructing the task, we randomly designate which political party has the veto power.

**Political Goals.** Political goals indicate when the political consensus is achieved. In *PoliCon*, we define three different political goals as follows: (1) **Passing a Resolution**: This is the most common political consensus objective, aimed at finding a consensus resolution that can be passed under a specific power structure and voting mechanism detailed above. (2) **Rawlsianism**: To study the extent to which LLMs can accommodate the interests of minority groups, following the Rawlsian principle (Rawls, 2017), the political goal in this context is to formulate a resolution that maximizes the benefits for the party with the least benefits. In this setting, $u = \min_{i \in n}(u_i)$. (3) **Utilitarianism**: Following the Utilitarian principle (Mill, 2016), the political goal is to formulate a resolution that maximizes the sum of benefits for all parties. Under this setting, $u = \sum_{i=1}^{n} u_i$.

It is worth noting that, in our defined political goals, only the passing resolution setting requires different voting mechanisms and corresponding power structures, which return a boolean variable indicating whether a vote passes. For Rawlsianism and Utilitarianism, only the corresponding voting score needs to be considered. Therefore, by combining different power structures, voting mechanisms, and political goals, we establish five distinct settings: Passing Simple Majority (SM), Passing Two-Thirds Majority (2/3M), Passing Veto Power (VP), Rawlsianism (Rawls), and Utilitarianism (Util). These can further be combined with three party number configurations (2, 4, or 6 parties), resulting in 15 task settings. Since each data record we collected represents an independent political issue, our framework can construct 28,620 distinct political scenarios altogether.

## 4 EXPERIMENTS

In this section, we use *PoliCon* to conduct comprehensive experiments to evaluate six current representative LLMs, specifically on their capability to achieve diverse objectives of political consensus. We selected two close-sourced models: GPT-4o (Hurst et al., 2024) and Gemini-2.5-Flash (Gemini-2.5) (DeepMind, 2024), as well as four open-sourced models from different vendors and with varying parameters: Qwen2.5-32B-Instruct (Qwen2.5-32B), Qwen2.5-72B-Instruct (Qwen2.5-72B) (Yang et al., 2024; Team, 2024), Llama-3.3-70B-Instruct (Llama-3.3-70B) (AI@Meta, 2024), and 671-billion-parameter DeepSeek-V3.1 (DeepSeek-AI, 2024). All LLMs are set up with standardized inference settings, including a temperature of 0.7 and top-p sampling of 0.95. For Gemini-2.5 and DeepSeek-V3.1, we use their thinking versions.

To clearly demonstrate the model's capabilities, we also introduce two rule-based baselines: Random and Greedy. The Random method randomly selects the stance of a party as the final resolution, while the Greedy method selects the stance of the party with the largest number of seats as the resolution.

In the following subsections, we will demonstrate how *PoliCon* can be used to investigate these four questions: (1) Can our evaluator simulate the voting results well? (2) How does the performance differ among various LLMs and (3) task settings and issue topics? (4) Do the tested LLMs tend to have biases towards the political parties?

### 4.1 CAN OUR EVALUATOR SIMULATE THE VOTING RESULTS WELL?

As mentioned in subsection 3.1, to construct our open-ended evaluation for assessing whether the tested LLM can reach the objectives of each political consensus, we first need a sub-evaluation module to score the preferences of each stakeholder. To this end, we introduce a GPT-4o-mini-backboned evaluator and request it to output an integer scalar between 0 and 9 to simulate the percentage of MEPs from each party who voted in favor. For evaluating its reliability, we will introduce it from two perspectives in the following: the consistency with real-world ground truth voting results and the consistency with human annotators.

**Consistency with real-world voting results.** For this experiment, we randomly sample 100 issues for every combination of party and topic in each parliamentary term, resulting in approximately 41,800 testing samples in total. We conduct a consistency validation experiment by comparing our evaluator's scores with the real-world party voting results, where we achieve a high Pearson correlation coefficient of 0.83.

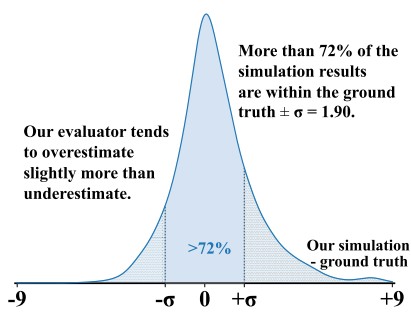

Additionally, referring to existing work (Zhou et al., 2023), we plot Figure 4 to further illustrate the distribution of computational errors for our evaluator. The error is calculated by subtracting the ground truth voting score from the simulated score of our evaluator. The detailed calculation process is shown in the Appendix E.1. It can be observed that the majority of the simulation results (>72%) are centered within the standard deviation $\sigma$ ($\pm1.90$) around the real voting results, with more simulation results showing a slight overestimation than underestimation. Based on

Figure 4: The error distribution between our simulation and the ground truth voting results. The x-axis indicates the difference between the evaluator's simulation results and the ground truth.

the above experiments, we answered the question raised in the caption of the subsection with our evaluator is sufficiently capable of simulating each party's voting results for the current resolution. More detailed experimental results can be found in Appendix F.1.

**Consistency with human annotators.** In the above experiments, we demonstrate that our evaluator maintains a high level of agreement with real-world ground-truth voting results. However, to further verify that our evaluator can generalize to assessing consensus decisions generated by LLMs, we conduct an additional human study to examine whether it can maintain consistency with human annotators.

Because our task involves both AI and political science, we recruit 20 human annotators. Half of them have AI-related backgrounds, and the other half are law students, allowing the team to cover a broader range of relevant domain knowledge.

Table 1: Performance of different LLMs on *PoliCon*. The values in square brackets indicate the range of each metric, and all metrics follow the principle that higher values are better. The background color of the table cells deepens as the performance improves. The blue color scheme represents metrics in the 0-1 range, while the red color scheme represents metrics in the 0-9 range.

| Model | SM [0-1] ↑ | | | 2/3M [0-1] ↑ | | | VP [0-1] ↑ | | | Rawls [0-9] ↑ | | | Util [0-9] ↑ | | |
|---|---|---|---|---|---|---|---|---|---|---|---|---|---|---|---|
| | 2 | 4 | 6 | 2 | 4 | 6 | 2 | 4 | 6 | 2 | 4 | 6 | 2 | 4 | 6 |
| Random | 0.56 | 0.53 | 0.56 | 0.29 | 0.20 | 0.14 | 0.36 | 0.35 | 0.38 | 2.59 | 2.01 | 1.77 | 5.04 | 4.78 | 4.80 |
| Greedy | 0.80 | 0.74 | 0.73 | 0.45 | 0.37 | 0.28 | 0.46 | 0.44 | 0.44 | 2.61 | 2.02 | 1.74 | 5.07 | 4.79 | 4.79 |
| Qwen2.5-32B | 0.74 | 0.80 | 0.87 | 0.34 | 0.39 | 0.40 | 0.47 | 0.55 | 0.62 | 4.02 | 3.50 | 3.19 | 6.01 | 6.27 | 6.38 |
| Llama-3.3-70B | 0.72 | 0.78 | 0.86 | 0.37 | 0.45 | 0.48 | 0.46 | 0.55 | 0.63 | 3.98 | 3.42 | 3.11 | 6.08 | 6.40 | 6.56 |
| Qwen2.5-72B | 0.76 | 0.82 | 0.88 | 0.40 | 0.47 | 0.49 | 0.50 | 0.57 | 0.65 | 4.11 | 3.46 | 3.13 | 6.11 | 6.39 | 6.53 |
| GPT-4o | 0.83 | 0.87 | **0.92** | 0.51 | **0.57** | **0.63** | 0.54 | 0.62 | 0.69 | 4.50 | 3.80 | 3.42 | **6.40** | **6.62** | **6.80** |
| Deepseek-V3.1 | 0.87 | 0.89 | **0.93** | 0.52 | **0.57** | **0.63** | 0.58 | 0.64 | **0.71** | 4.52 | 3.78 | 3.42 | 6.38 | **6.62** | 6.77 |
| Gemini-2.5 | **0.88** | **0.90** | 0.90 | **0.53** | **0.57** | 0.58 | **0.61** | **0.66** | 0.70 | **4.60** | **3.91** | **3.51** | 6.39 | 6.56 | 6.68 |

We further design and create a refined and user-friendly user interface (UI) to make the annotation process easier for human annotators. The page includes the following design elements: first, it presents the annotators with the issue title, relevant background information, party stances, and the corresponding generated resolution. Then, the annotators are asked to assign scores for the consistency between the resolution and the stances, as well as for the feasibility of the resolution. In the scoring section, we provide detailed criteria that explain the standards for each score range. Meanwhile, to accommodate annotators with different native languages, we also implemented a multilingual switching mode. The detailed UI design can be found in Appendix E.2.

We randomly sample 100 consensus resolutions generated by LLMs for evaluation. To ensure annotation robustness, each data item is assigned to three different annotators, yielding a total of 300 annotation results. Consistent with the evaluator's scoring method, we use the expected values of consistency and feasibility as the final evaluation scores. As shown in Table 2, we present the consistency results of our evaluator compared with the ground truth and human evaluations. We report the mean error, the standard deviation $\sigma$, and the proportion of samples whose errors are within $\pm\sigma$.

Table 2: Consistency of our evaluator with ground truth and human evaluations.

| | Ground truth | Human eval. |
|---|---|---|
| Mean error | 1.36 | 1.61 |
| $\sigma$ | 1.90 | 1.92 |
| Within $\pm\sigma$ | 72% | 72% |

It can be seen that the mean error between our evaluator and human annotations is only 1.61, and more than 72% of the data errors are within $\pm 1.92$, which demonstrates that our evaluator is sufficient to generalize in evaluating LLM-generated consensus resolutions. Another interesting phenomenon is that the consistency between our evaluator and human annotators is highly aligned with the consistency between our evaluator and the real-world ground-truth voting results, indicating that our consistency-computation method based on real voting data is also effective.

## 4.2 PERFORMANCE ANALYSIS FOR VARIOUS LLMS

We utilize the *PoliCon* evaluation framework described in subsection 4.1 to assess the performance of six LLMs on *PoliCon*. The results are depicted in Table 1, which presents the average scores across all our 15 task settings described in subsection 3.2. For the SM, 2/3M, and VP, the scores represent the average passing rates ranging from 0 to 1. For Rawls and Util, the scores represent the average results obtained from the corresponding calculation methods, ranging from 0 to 9. All these metrics are higher-the-better.

We find that **Gemini-2.5** performs the best, achieving the best results on 60% of the tasks. Deepseek-V3.1 and GPT-4o follow with both attaining top performance on 33% of the tasks. We also compare the performance differences among other evaluated LLMs and identify the following trends: (1) Thinking models like Gemini-2.5 and Deepseek-V3.1 generally outperform no-thinking models like GPT-4o and Llama-3.3-70B. (2) Commercial models typically outperform non-commercial models. (3) Based on the results of four open-sourced models with known parameter sizes, we find that the performance is generally positively correlated with the model size.

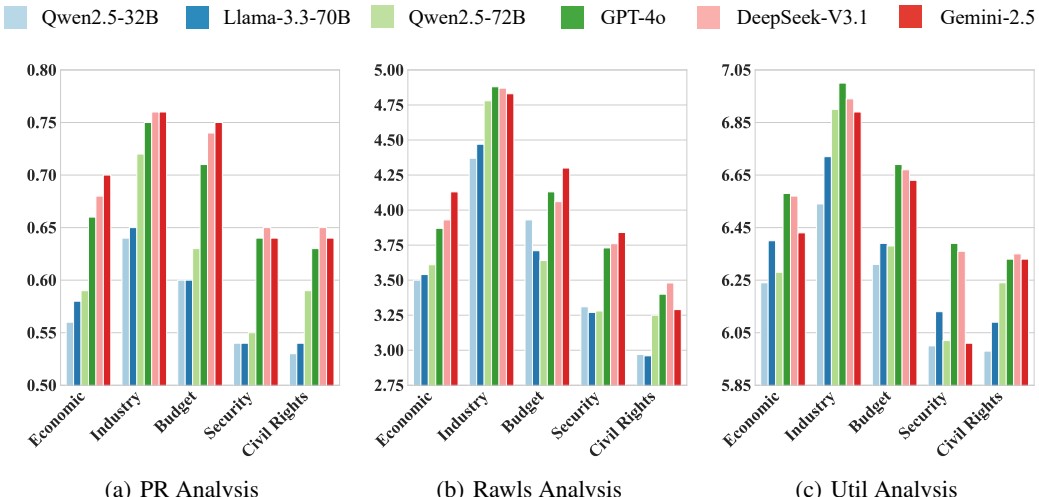

(a) PR Analysis    (b) Rawls Analysis    (c) Util Analysis

Figure 6: The average results of the six evaluated LLMs of the five coarse-grained topics on passing resolution (PR, including SM, 2/3M, and VP), Rawls, and Util political goals.

We further report results for two rule-based baselines, Random and Greedy. Random attains the lowest scores on all tasks, indicating that current LLMs have a nontrivial advantage over naive random choice in political consensus finding. Greedy generally outperforms Random but remains below LLM-generated consensus texts, except in the two-party passing resolution settings. In the SM and 2/3M settings, it surpasses Qwen2.5-32B, Llama-3.3-70B, and Qwen2.5-72B and ranks behind GPT-4o, and in the VP setting, it is comparable to Llama-3.3-70B. These results show that, under these conditions, most open-source models still retain substantial room for improvement.

Additionally, we analyze whether a common strategy exists for LLMs to achieve political consensus under various power structures, excluding two-party scenarios. As shown in Figure 5, we find that under both simple majority and two-thirds majority systems, LLMs lack the ability to unite smaller parties to achieve collective welfare. Instead, successful proposals often rely on the support of the largest party, indicating that the votes of dominant parties are foundational for approval in most cases.

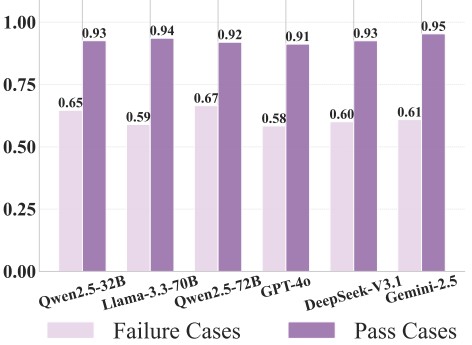

Figure 5: The average contribution ratio of the largest party to other parties in failed and passed cases across SM and 2/3M.

## 4.3 PERFORMANCE ANALYSIS FOR DIFFERENT TASK SETTINGS AND ISSUE TOPICS

In this section, we demonstrate how different task settings and issue topics in *PoliCon* influence LLMs' ability to find political consensus, which are presented separately as follows:

**Analysis for Different Task Settings.** As shown in Table 1, for the political goal of passing a resolution, SM is the simplest, and most models can perform well. However, in the 2/3M and VP settings, model performance declines significantly, indicating that the capabilities of existing LLMs generally lie in the gap between the increased difficulty of SM and these two settings. We further find that as the number of parties increases, the results of most models gradually rise. This could be due to our task construction prioritizing parties with the most diverse positions, complicating reconciliation with fewer parties. For the Rawls objective, however, the success rate of models decreases as the party number increases. This aligns with the task's definition, as the more participants there are, the harder it becomes to avoid neglecting any party's interests, presenting a significant challenge for current LLMs in this task.

**Analysis for Different Issue Topics.** As shown in Figure 6, we analyze the experimental results of five coarse-grained topics. These results suggest that the difficulty of different topics shows certain

Table 3: Scores of different LLMs regarding the degree of bias between political parties.

| Qwen2.5-32B | Llama-3.3-70B | Qwen2.5-72B | GPT-4o | Deepseek-V3.1 | Gemini-2.5 |
|---|---|---|---|---|---|
| 2.74 | 3.20 | 3.05 | 2.79 | 2.64 | 2.34 |

Figure 7: Partisan bias of the tested LLMs. (Top) Average scores from the tested LLMs on different parties. (Middle) Ground truth votes of different parties. (Bottom) Scores of random assignment.

similarities across various parliamentary settings. Specifically, topics involving policies, such as Security and Civil Rights, tend to be more challenging than those related to industrial development. This may be because these topics tend to present more complex and conflicting positions, requiring the evaluated LLM to possess stronger reasoning capabilities. For the complete experimental results of each fine-grained topic, see Appendix F.3.

Our experimental results successfully reveal the limitations of the current LLMs in political consensus finding. Although top-performing models like Deepseek-V3.1 and Gemini-2.5 achieve a success rate of 87-93% in SM scenarios, their performance significantly drops when faced with stricter consensus requirements. In 2/3M tasks, the success rate falls to 52-63%, and in the more challenging Rawls setting, it ranges from only 3.42-4.60. Additionally, when dealing with more complex topics such as security, these models still face considerable challenges.

### 4.4 BIAS EVALUATION FOR THE TESTED LLMS

We further investigate the partisan bias of tested models. As surprisingly shown in Figure 7, though our party seats were randomly reassigned, the scores across different parties still resemble the distribution of real-world voting results. This indicates that the tested models are somehow influenced by the real-world party preferences. In contrast, the score distribution for random resolutions is entirely different, suggesting that this bias is not caused by our evaluator or the data cleaning process.

In Table 3, we calculated the variance of scores across different terms of parties for each model. We found that, generally, as this variance decreases, the corresponding performance in Table 1 increases. This makes sense because when models discard their bias towards political parties, they can better adapt to the party weights in our setting and produce reasonable resolutions.

## 5 CONCLUSION

In this work, we introduced *PoliCon*, a novel benchmark constructed from 2,225 European Parliament deliberation records to evaluate LLMs' ability to achieve diverse political consensus objectives across diverse real-world collective decision-making settings. Our framework incorporates key factors like political issues, goals, party stances, and power structures, with an evaluation system that first simulates real voting outcomes and then assesses whether relevant political consensus can be achieved based on the social choice theory. Our experiments highlight *PoliCon*'s promise as an effective platform for studying LLMs' ability to promote political consensus. To our knowledge, *PoliCon* represents the first comprehensive benchmark for assessing diverse political consensus achieving capabilities in LLMs, offering both a valuable evaluation platform and new insights into how LLMs navigate complex governance scenarios. We believe our work points to one of the important directions for the intersection research between AI and political science.

ETHICS STATEMENT

The *PoliCon* benchmark is constructed from openly available sources, including the European Parliament website, HowTheyVote, and the VoteWatch Europe dataset. Both the official website of the European Parliament and HowTheyVote allow the use of their data as long as the source is cited, while the VoteWatch Europe dataset follows the CC 4.0 license. Importantly, while *PoliCon* is built upon authentic deliberation records, it does not reflect or predict the actual positions of the European Parliament, but rather serves as a research framework. Given the potential societal risks of applying AI in governance, such as reinforcing systemic biases, creating ideological echo chambers, fostering over-reliance on automated decision-making systems, and amplifying politically sensitive or divisive content, we urge all users of this benchmark to be acutely aware of these risks and to proceed with a high degree of caution and ethical responsibility. To further mitigate risks of potential misuse, we will release *PoliCon* under a research-only license agreement. The authors bear no responsibility for misuse or politically motivated interpretations. A more detailed ethical statement is provided in Appendix H.

REPRODUCIBILITY STATEMENT

We provide comprehensive details to ensure reproducibility: the construction of the benchmark is described in section 3, including data collection, cleaning, and task design; the full implementation of our experiments is presented in section 4, covering the models under evaluation, their configurations, and our evaluation procedure. All implementation details and experimental settings required for reproduction are included in the paper and the appendix.

ACKNOWLEDGMENTS

The work of PoliCon was sponsored by the National Natural Science Foundation of China (62376031, 62376013, 623B2003, 624B100026). This work was also supported by the Natural Science Foundation of Beijing (QY24041). Any opinions, findings, or conclusions expressed in this work do not necessarily reflect the views of the funding agencies.

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

# Supplementary Material

## Table of Contents

# A  DATASET CONSTRUCTION DETAILS

In this section, we will provide a detailed explanation of the complete process of data collection and post-processing mentioned in subsection 3.1.

## A.1  DATA COLLECTION PROCESS

In this subsection, we will focus on the perspective of large-scale data crawling, introducing the methodology and process of raw data collection.

### A.1.1  DATA SOURCES

The data collection process for the *PoliCon* begins with the VoteWatch Europe dataset (HIX et al., 2022), which contains structured voting records of the European Parliament (EP) spanning 18 years from 2004 to 2022. Since the data for the five years from 2004 to 2009 is incomplete, we have excluded it. The portion of the dataset we use includes: (1) Excel files with metadata for the seventh, eighth, and half of the ninth European Parliament terms (2014-2022), including vote identifiers, titles, issue topics, etc.; (2) Roll call voting records mapping MEPs to vote outcomes, including six categories: in favor, against, abstain, absent, not voted, not an MEP; (3) URLs of the original sources from the official website of the European Parliament regarding where to obtain the voting data.

The second data source to be introduced is HowTheyVote[3]. This data source also presents roll call voting data for each MEP and provides URLs that link to the data sources. There are two main differences between this data source and the VoteWatch Europe dataset: first, it only includes data from the 9th and 10th European Parliament sessions after 2019. Second, it contains URLs for both the voting data and related records of resolutions and debates from the European Parliament's official website.

The last and most important data source is the European Parliament's official website[4]. This source lacks systematic organization of roll call voting data for each resolution (it is not absent, but it is not easy to scrape on a large scale, which makes us rely on other data sources for voting record extraction). However, it provides extensive and detailed data on resolutions and debate records for each decision.

Through HowTheyVote, we discovered how voting URLs can correspond to their respective resolution and debate records via specific web navigation. Once this information is obtained, we can cleverly combine the voting information from the VoteWatch Europe dataset and HowTheyVote, along with the voting source URLs from the European Parliament's official website, to access the resolution and debate record data corresponding to each decision. This establishes the foundation for large-scale data scraping.

### A.1.2  UNIFIED URL PARSING

On the official website of the European Parliament, some URLs have multiple redirect issues, which means the directly indexed webpage is not the original record's page. To solve this problem, we developed an automated pipeline to handle specific short URL issues in the European Parliament system, which consists of the following key steps: First, we sent HTTP HEAD requests for all short URLs (in formats such as `europarl.europa.eu/doceo/xxx`) to fully trace redirection chains. Second, the final URLs were validated against an official domain whitelist to ensure that all resolved results point to valid European Parliament resources. Finally, cryptographic hashing was employed for integrity verification, storing both original and resolved URLs while generating SHA-256 digests for audit trails.

This solution effectively addresses URL standardization issues in the European Parliament's official document system while preserving complete data provenance information. By combining the verification of the network protocol layer with cryptographic validation, a dual guarantee mechanism was established. In this way, we can ensure that every URL can index the corresponding webpage information.

---

[3]https://howtheyvote.eu
[4]https://www.europarl.europa.eu

### A.1.3 WEB CONTENT EXTRACTION

We employed the Python `BeautifulSoup` library[5] to parse the raw HTML content from the official European Parliament website. However, the European Parliament's web pages do not follow a uniform HTML format, especially those targeting paragraphs with distinct stylistic features (such as those with `margin-left:17.85pt` formatting). This necessitates handling these diverse special webpage structures during the data scraping process to accurately capture the resolution body text. To address this situation, we performed customized processing for each special case of uniquely occurring resolution webpage format, including identification methods like paragraph filtering using standard resolution startings (e.g., "The European Parliament"), ultimately obtaining complete raw resolution data.

For debate records, through document object model (DOM) tree traversal techniques, we identified HTML elements containing debate records (nodes with the `doceo-ring-steps-step-label` class). During speech content extraction, the system automatically filters procedural statements (e.g., chairperson remarks like "The President") while retaining substantive policy debate content. This process combines dual verification mechanisms of semantic analysis and rule-based pattern matching.

For the special requirements of the 9th European Parliament (2019 - 2022), we developed a parsing adapter based on URL path heuristic rules. By recognizing specific path patterns (such as URLs containing `/A8/` or `/B9/` identifiers), the system can automatically switch the corresponding content extraction strategies to effectively address technical challenges caused by structural changes in the websites of parliament. The framework supports dynamic loading of new parsing rules, ensuring long-term system maintainability. Key features of this implementation include: (1) Context-aware parsing for different parliamentary terms; (2) Automated detection of document structural changes; and (3) Fallback mechanisms for handling legacy formats.

These approaches leverage the standardized typography of European parliamentary document systems to reliably extract structured textual content. In this way, we obtained the 30,698 original parliamentary deliberation records mentioned in subsection 3.1.

### A.1.4 REDUNDANT DATA FILTERING

In the European Parliament, each issue requires careful consideration before reaching a final resolution, so clearly, no resolution can be finalized in just one meeting. As a result, the parliamentary records show that each issue typically undergoes more than ten rounds of revisions and voting. Therefore, we need to efficiently eliminate the intermediate processes of these issues, leaving only the final effective data version. To address this problem, we implemented a rigorous two-phase deduplication mechanism to ensure the uniqueness and authority of legislative data. The first phase handles duplication at the legislative level, while the second phase resolves document-level ambiguities.

**Legislative Level Uniqueness Guarantee.** From the perspective of the procedural legitimacy of the European Parliament, the final decision should be based on the roll-call vote results of the final vote[6]. This information is represented in the VoteWatch Europe dataset with the label `final_vote=1`. Therefore, in this paper, we only retain the voting records with this label for each issue.

**Document-Level Disambiguation.** When multiple entries referencing identical legislative content (identified by URL matching) were detected, we adopted the most recent-first principle, retaining the latest record according to the `vote_timestamp` field. This mechanism establishes a bijective relationship between legislative acts and their canonical representations while maintaining the temporal logic of data updates.

### A.2 VOTE IN FAVOR CALCULATION

In the VoteWatch Europe dataset, all roll-call voting data records the voting decisions of each MEP and uses the following six labels to record their voting outcomes: in favor, against, abstain, absent, not voted, and not an MEP. For the *PoliCon* setup, we need the proportion of MEPs voting in favor of

---

[5]https://pypi.org/project/beautifulsoup4
[6]https://www.europarl.europa.eu/doceo/document/RULES-10-2025-01-20-RULE-047_EN.html

each party on each issue. Therefore, we need to further process the voting data. First, we need to match each MEP to their respective party, which is labeled in the VoteWatch Europe dataset. However, the names of the same parties are not consistent (due to different names and typos), so we reclassified these to accurately identify the party each MEP belongs to. We then calculated the proportion of MEPs voting in favor of each party on each issue. It's important to note that, as mentioned above, there are six voting outcome labels, but we only use the "in favor" label to calculate the proportion of votes in favor. As for the HowTheyVote data source, the proportion of votes in favor of each party is already calculated, so for the ninth parliament, we don't need to perform this operation.

## A.3 USED POLITICAL GROUP NAME ABBREVIATIONS

For each political party in the European Parliament, there are different names and abbreviations. For example, the European People's Party has official abbreviations like EPP and PPE, among other variations. Due to the different languages used in European Union countries, there are corresponding abbreviations for different languages as well. Therefore, in this document, we need to introduce the party name abbreviations used in *PoliCon* and their corresponding party names.

Table 4: Used political group name abbreviations in the 7th parliament.

| Abbreviation | Full Name |
|---|---|
| EPP | European People's Party |
| EFD | Europe of Freedom and Democracy |
| SD | Progressive Alliance of Socialists and Democrats |
| ALDE | Alliance of Liberals and Democrats for Europe Party |
| ECR | European Conservatives and Reformists Group |
| GREEN/EFA (GREEN_EFA in dataset) | The Greens/European Free Alliance |
| GUE/NGL (GUE_NGL in dataset) | The Left in the European Parliament |

Table 5: Used political group name abbreviations in the 8th parliament.

| Abbreviation | Full Name |
|---|---|
| EPP | European People's Party |
| SD | Progressive Alliance of Socialists and Democrats |
| ECR | European Conservatives and Reformists Group |
| EFDD | Europe of Freedom and Direct Democracy |
| GREEN/EFA (GREEN_EFA in dataset) | The Greens/European Free Alliance |
| GUE/NGL (GUE_NGL in dataset) | The Left in the European Parliament |
| ALDE | Alliance of Liberals and Democrats for Europe Party |
| ENF | Europe of Nations and Freedom |

Table 6: Used political group name abbreviations in the 9th parliament.

| Abbreviation | Full Name |
|---|---|
| EPP | European People's Party |
| SD | Progressive Alliance of Socialists and Democrats |
| ECR | European Conservatives and Reformists Group |
| RENEW | Renew Europe |
| GREEN/EFA (GREEN_EFA in dataset) | The Greens/European Free Alliance |
| GUE/NGL (GUE_NGL in dataset) | The Left in the European Parliament |
| ID | Identity and Democracy |

In the 7th parliament term, the party abbreviations we used were EPP, EFD, SD, ALDE, ECR, GREEN/EFA, and GUE/NGL, as shown in Table 4. Interestingly, the abbreviation GUE/NGL for The Left in the European Parliament does not directly correspond to its full English name. This is because the party was originally formed by the merger of the Confederal Group of the European United Left (GUE) and the Nordic Green Left Alliance (NGL). Information on party abbreviations for the 8th and 9th parliaments is shown in Table 5 and Table 6.

## A.4 OUTPUT DATA ENTRY SCHEMA

Finally, after the large-scale crawling and preprocessing steps described above, we obtained 2,225 high-quality complete parliamentary record data entries. For each entry, we used the following JSON format for storage:

```
{
  "excel_title": "Issue Title",
  "web_title": "HTML-Derived Title",
  "topic_select": "Fine-grained Topic Name",
  "text_url": "Canonical Document URL",
  "resolution": "Full Resolution Text",
  "votes_total": {"FOR": 75, "AGAINST": 124, ...},
  "votes": [
  {
    "group": {"code": "EPP", "label": "...", ...},
    "stats": {"FOR": 35, "AGAINST": 72, ...}
  }, ...
  ],
  "debate": {
    "title": "Debate Transcript Title",
    "views": [{"speaker": "MEP Name", "debate": "Utterance"}, ...]
  }
}
```

The JSON file contains all the quintuple raw information mentioned in subsection 3.1, namely issue, topic, debates, resolution, and votes. We will introduce which keys in the JSON field correspond to these raw pieces of information as follows: (1) issue: excel_title and web_title provide the official and HTML-derived issue titles, respectively. We use "excel_title: web_title" as the issue's final name; (2) topic: top_select indicates the policy area, and text_url links to the canonical document; (3) debates: The debate field describes the original debate record, where the title is the debate webpage's title, and views include the current speaker's name (speaker) and their speech content (debate); (4) resolution: Indicated by the resolution field; (5) votes: We have separately saved the results of two types of votes: the votes_total, which represents the overall votes for the resolution in the parliament, and the votes, which represents the votes of each party on the resolution. In the votes field, group indicates the information of the party currently voting, and stats represents the record of their votes.

## A.5 DATA FILTERING ANALYSIS

Our pipeline implemented rigorous quality controls across three parliamentary terms, with key metrics shown in Table 7.

Table 7: Data filtering ratio by different parliamentary terms.

| Metric | 7th | 8th | 9th |
|---|---|---|---|
| Initial Records | 6,963 | 10,276 | 13,459 |
| Duplicates Removed | 5,333 (76.6%) | 8,349 (81.2%) | 12,414 (92.2%) |
| Debate Transcripts Missing | 580/1,630 (35.6%) | 800/1,927 (41.5%) | 487/1,045 (46.6%) |
| Final Raw Valid Records | 1,050 (15.1%) | 1,127 (11.0%) | 558 (4.1%) |

**Initial Records.** The initial records represent the total number of unprocessed voting records collected from raw data sources. For instance, the 7th term had 6,963 records, while the 9th term saw a significant increase to 13,459 records, and that is only half of the term. This metric is significant as it reflects the original scale of data collection, illustrating a 93% growth from the 7th to the 9th term.

**Duplicates Removed.** Duplicates are identified through the process described in subsubsection A.1.4 and subsequently removed from the dataset. The key characteristics of this process include both absolute numbers (e.g., 8,349 removed in the 8th term) and percentages (81.2%). The duplication rate increases across terms, from 76.6% in the 7th term to 92.2% in the 9th term. Notably, the high duplication rate in the 9th term (92.2%) perhaps reflects the increased frequency of its discussion issues.

**Debate Transcripts Missing.** Some voting records lack corresponding parliamentary debate texts, resulting in missing debate transcripts. This issue is represented in two forms: as a numerator/denominator (e.g., the 7th term: 580/1,630) and as a percentage (ranging from 35.6% to 46.6%). There is a consistent upward trend in the missing rate, with the 9th term reaching 46.6%, indicating that nearly half of the records are devoid of contextual debate information.

**Final Raw Valid Records.** The final raw valid records are those that are available and pass all quality checks before the data cleaning process. They are calculated by subtracting duplicates and missing records from the initial records. For example, in the 7th term, the calculation is 6,963 (initial records) - 5,333 (duplicates) - 580 (missing records) = 1,050 valid records. Despite the initial growth of the records, the number of valid records in the 9th term (558) decreased by 11% compared to the 7th term (1,050), highlighting the decline in the usability of the data.

The above analysis reveals that the data we used in *PoliCon* only accounts for 7.2% of the original data, reflecting that the data we adopted consists of carefully selected high-quality deliberation records.

### A.6 DATA CLEANING DETAILS

Due to the redundancy of the raw data, such as the large number of useless remarks in the debate, after collecting the original data, we further used DeepSeek-R1 (Guo et al., 2025) and rule-based methods for data cleaning and post-processing operations. First, we used DeepSeek-R1 to reorganize the resolutions and remove redundant parts while retaining the original resolution format. We further summarized the background of the current parliamentary discussion topics based on issue, resolution, and debate information.

Next, we processed the voting data, where the original voting information included each member's vote on each issue. We matched each member with their parliamentary party and calculated the voting information for each party on the current resolution. We calculated the proportion of members within the party who voted in favor and rounded down to an integer between 0 and 9 as the party's preference score for the resolution.

Table 8: Paraphrase word list for the data post-processing procedure.

| Attitude | Word List |
|---|---|
| Support Verbs | support, agree, endorse, advocate, approve, sanction, uphold, accept, promote |
| Oppose Verbs | oppose, reject, disapprove, condemn, conflict, doubt, challenge, dispute, against |
| Support Adverbs | fully, totally, completely, absolutely, entirely, fundamentally, firmly |
| Oppose Adverbs | partly, slightly, partially, confitionally |

Subsequently, based on the resolution and each party's voting information, we let DeepSeek-R1 filter each party's stances on the issue from the debate data. If a party did not express a stance or opinion in the debate, we removed the party from the issue. The detailed prompt can be found in Appendix C.1. Then we used rule-based methods to perform synonym replacement on tone words expressing political party stances. For example, "strongly agree" can be replaced with "fully endorse" or "totally support", among others (detailed in Table 8). This approach increases data diversity and helps reduce

the bias in word choices introduced by the LLM. Additionally, since all stances in the debate data are related to the current committee proposal or submitted resolution, and we need the LLM to provide new resolutions when using this data, we replaced the word "resolution" in each party stance with the synonym "issue" to adjust the stances on the resolution to stances on the issue. This eliminates conflicts in referential terms between the new resolution generated by the tested LLMs and the word "resolution" in the stances during practical data usage.

We applied the process to each data entry. Through this approach, we cleaned the raw data into sextuples of (issue, topic, background, stances, resolution, votes), where stances and votes contain relevant information from all parties involved in the discussion of the issue.

After this data cleaning process, we once again inspected the data and carried out two filtering operations: (1) filtering out records with missing resolution or stances; and (2) classifying the final records by topic and removing topics with too few records. In the following paragraph, I will introduce these two steps separately:

**Filtering records with Missing Resolution or Stances.** For records where the resolution is an empty string, the handling is straightforward by directly removing. The situation is slightly more complex for missing stances, because sometimes the stances string is not empty but is None or contains fewer than five words. Therefore, we adopt a rule-based method to filter original data with missing stances, namely, checking whether it is None, whether its length is less than 50 characters, or whether it is an empty string. If all stances of a record are filtered out based on these rules, the entire record is removed. In this process, removing empty resolutions filtered out 99 records, while removing empty stances filtered out 375 records. Thus, a total of 99 + 375 = 474 records were filtered out in this step.

**Filtering Topics with Too Few records.** In this process, we first categorize all records according to their topics. Since some topics contain very few records, for example, the "juridical affairs" topic includes only 13 records, treating them as independent tasks would provide insufficient data. Therefore, we remove topics with 25 or fewer records. In this step, two topics are removed: "juridical affairs" (only 13 records) and "petitions" (only 23 records), resulting in a total of 36 filtered records.

Therefore, in these final two filtering steps, we filtered a total of $474 + 36 = 510$ records. After these two steps, the total number of records decreased from the original 2,735 to the final 2,225 complete ones.

Table 9: Overview of the fine-grained topics and their contents (with some topic names abbreviated for convenience in the table).

| Topic Name | Detailed Content |
| --- | --- |
| Agriculture | Agricultural policy, rural development, and food security. |
| Budget | Budget negotiations, annual budget adoption, and financial reforms. |
| Budgetary Control | Budget implementation, ensures financial transparency, combats fraud, and promotes accountability. |
| Civil Liberties | Policies on civil liberties, justice, and home affairs, focusing on fundamental rights, migration, data protection, and security. |
| Constitutional Affairs | Constitutional affairs, focusing on treaty implementation, institutional reforms, and democratic governance . |
| Culture & Education | Policies on culture, education, media, youth, and sports, managing flagship programs to promote cultural diversity, education, and cross-border cooperation. |
| Development | Global sustainable development, overseeing EU aid budgets, combating poverty, and strengthening partnerships to tackle inequality and humanitarian challenges. |
| Economic Affairs | Regulation of financial services, the free movement of capital, payments, taxation, competition policies, and the international financial system. |
| Employment | Employment policies, workers' rights, social inclusion, and addressing challenges like economic transitions and inequality through legislative oversight. |
| Public Health | Environmental policies, climate action, and food safety, prioritizing Green Deal implementation, biodiversity, and sustainable transition, public health issues, including pharmaceutical reforms, disease prevention (e.g., cancer, mental health), health data governance, and reducing EU health inequalities. |
| Fisheries | Sustainable fisheries management, marine conservation, and socio-economic support for coastal communities under the Common Fisheries Policy reform. |
| Foreign & Security | Common Foreign and Security Policy and international agreements, defense strategies, hybrid threats, and military resilience in response to security challenges like Russia's war in Ukraine. |
| Gender Equality | Gender equality, combats violence/discrimination, and ensures women's inclusion in decision-making to address democratic deficits and societal fairness. |
| Industry & Energy | Legislation for energy transition, industry competitiveness, research innovation, digital/telecom policies, cybersecurity, and space policy to drive sustainable prosperity and EU strategic autonomy. |
| Internal Market | Single market rules, including digital integration and consumer protection, aiming to align with Green Deal objectives and high social/environmental standards. |
| International Trade | International trade agreements, WTO compliance, and scrutiny of trade policy implementation to strengthen the EU's global economic role. |
| Legal Affairs | Legal affairs, corporate law, intellectual property, and EU law simplification while ensuring institutional compliance and judicial oversight. |
| Regional Development | Cohesion policy, regional development, and solidarity through structural funds and multilevel governance to address disparities and future enlargement challenges. |
| Transport & Tourism | Transport/tourism decarbonization, digital transformation (e.g., autonomous vehicles), and sustainable mobility to meet climate goals and social equity. |

## B  TASK DETAILS

In this section, we will present the definitions of the coarse-grained and fine-grained topics we have categorized for each issue mentioned in subsection 3.1, as well as a more detailed display of the distribution of various political parties' stances in the semantic space.

## B.1 TOPIC CONTENTS

We categorize all collected data based on the topics outlined in the VoteWatch Europe dataset, which are derived from the committees of the European Union[7]. These 19 topics are then grouped into 5 coarse-grained categories:

**Economics.** Focuses on macroeconomic strategies. The fine-grained topics in this category are International Trade[8], Internal Market & Consumer Protection[9], Employment & Social Affairs[10], and Economic & Monetary Affairs[11].

**Industry.** Covers policies for specific industries. The fine-grained topics in this category are Agriculture[12], Fisheries[13], Transport & Tourism[14], and Industry, Research & Energy[15].

**Budget.** Encompasses budget policies for development. The fine-grained topics in this category are Development[16], Regional Development[17], Budget[18], and Budgetary Control[19].

**Security.** Addresses basic security guarantees, including military and health aspects. The fine-grained topics in this category are Environment & Public Health[20][21], and Foreign & Security Policy[22][23].

**Civil Rights.** Pertains to political and cultural issues. The fine-grained topics in this category are Culture & Education[24], Gender Equality[25], Civil Liberties, Justice & Home Affairs[26], Constitutional and Inter-institutional Affairs[27], and Legal Affairs[28].

We provide an overview of the main content covered under each topic in Table 9.

## B.2 MORE STANCES SEMATIC REPRESENTATION RESULTS

In subsection 3.2, we have previously provided a rough overview of the diversity of stances between parties in each parliamentary session. For illustration simplicity, we only displayed the distribution of 200 sampled data points in the semantic space for each party in the seventh and eighth parliaments. In this section, we will present a more detailed analysis of the sample data distribution and the complete data distribution for each party in every parliamentary session of *PoliCon*. This will further reveal the significant semantic diversity and stance conflicts between parties in *PoliCon*.

---

[7]https://www.europarl.europa.eu/committees/en/about/list-of-committees
[8]https://www.europarl.europa.eu/committees/en/inta/about
[9]https://www.europarl.europa.eu/committees/en/imco/about
[10]https://www.europarl.europa.eu/committees/en/empl/about
[11]https://www.europarl.europa.eu/committees/en/econ/about
[12]https://www.europarl.europa.eu/committees/en/agri/about
[13]https://www.europarl.europa.eu/committees/en/pech/about
[14]https://www.europarl.europa.eu/committees/en/tran/about
[15]https://www.europarl.europa.eu/committees/en/itre/about
[16]https://www.europarl.europa.eu/committees/en/deve/about
[17]https://www.europarl.europa.eu/committees/en/regi/about
[18]https://www.europarl.europa.eu/committees/en/budg/about
[19]https://www.europarl.europa.eu/committees/en/cont/about
[20]https://www.europarl.europa.eu/committees/en/envi/about
[21]https://www.europarl.europa.eu/committees/en/sant/about
[22]https://www.europarl.europa.eu/committees/en/afet/about
[23]https://www.europarl.europa.eu/committees/en/sede/about
[24]https://www.europarl.europa.eu/committees/en/cult/about
[25]https://www.europarl.europa.eu/committees/en/femm/about
[26]https://www.europarl.europa.eu/committees/en/libe/about
[27]https://www.europarl.europa.eu/committees/en/afco/about
[28]https://www.europarl.europa.eu/committees/en/juri/about

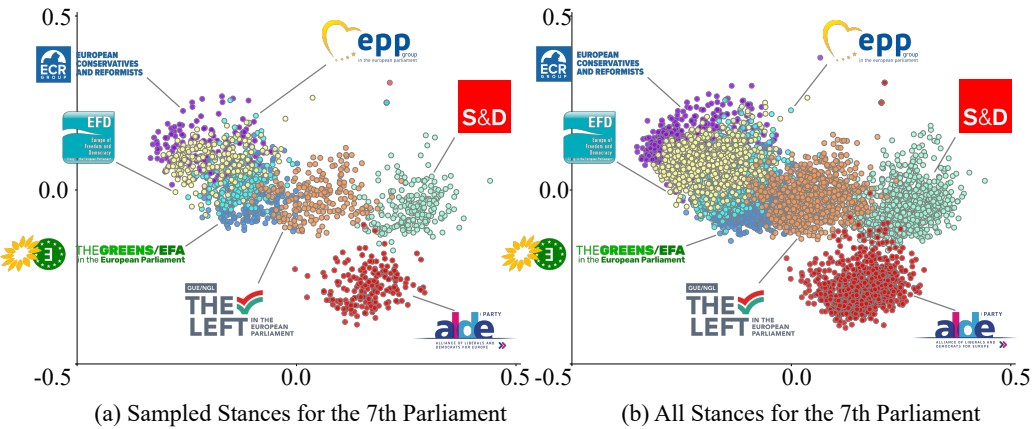

Figure 8: Semantic representation distribution of party stances (indicated by their symbols) in the 7th (2009-2014) term of the European Parliament in *PoliCon*. Figure (a) shows the sampled stances while Figure (b) illustrates all the stances.

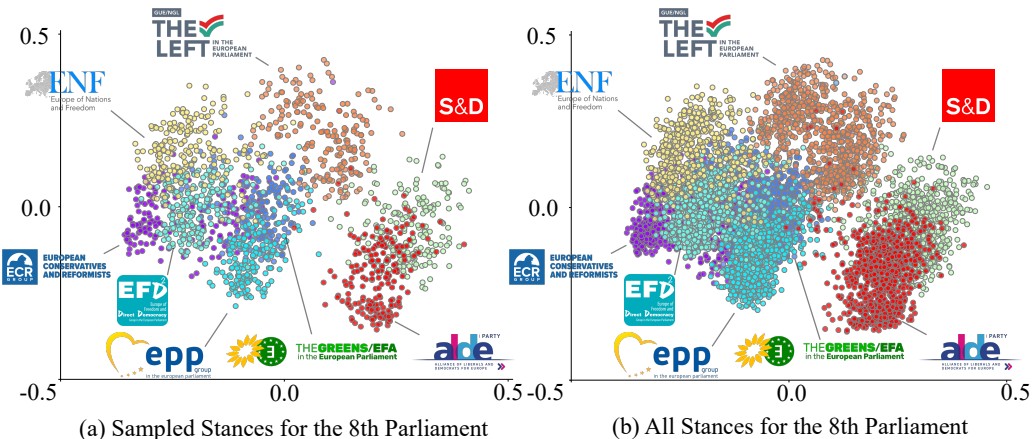

Figure 9: Semantic representation distribution of party stances (indicated by their symbols) in the 8th (2014-2019) term of the European Parliament in *PoliCon*. Figure (a) shows the sampled stances while Figure (b) illustrates all the stances.

As shown in Figure 8, Figure 9, and Figure 10, we present the sampled stances and all stances of all political parties during the 7th, 8th, and 9th terms of the parliament. From these three figures, it can be observed that the distribution results after sampling 200 data points for each party closely resemble those of the entire dataset, providing a strong reference value. Additionally, we can see that the distribution of party stances in the 7th and 8th terms of the European Parliament is more diverse compared to the 9th term. This may be due to factors such as Brexit (Besselink et al., 2019) and the rise of right-wing forces (Mudde, 2019; Servent, 2019; Abou-Chadi & Wagner, 2021), which highlights that our data analysis aligns with actual political trends.

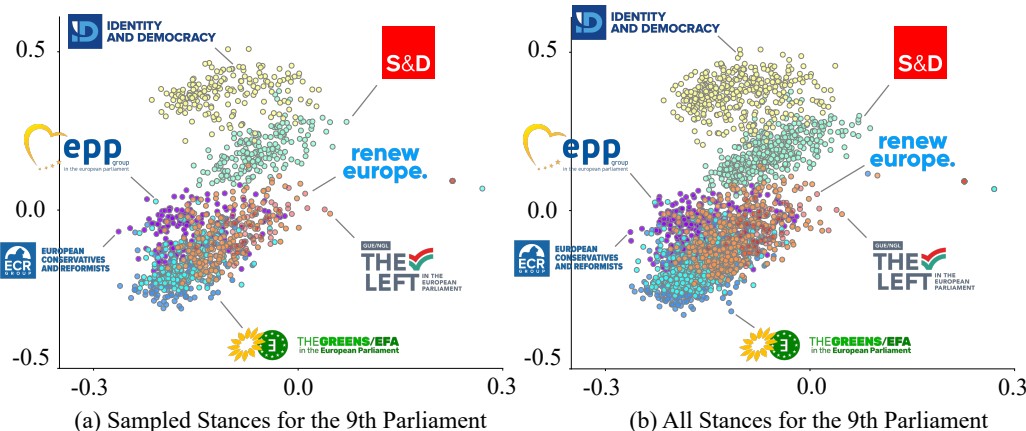

(a) Sampled Stances for the 9th Parliament     (b) All Stances for the 9th Parliament

Figure 10: Semantic representation distribution of party stances (indicated by their symbols) in the half of the 9th (2019-2022) term of the European Parliament in *PoliCon*. Figure (a) shows the sampled stances while Figure (b) illustrates all the stances.

## C   PROMPT DETAILS

In this section, we will demonstrate the details of all the prompts involved in this paper.

### C.1   DATA CLEANING PROMPTS

First, we will introduce the prompts used in the data cleaning and post-processing process. In this process, the prompts required include those for obtaining resolution, background, and extracting stances from the debate data of each party's MEPs. We will explain each of these in detail below.

> Summarize the key points of this European Parliament resolution in one continuous paragraph, without any formatting or line breaks. Begin the summary with 'The European Parliament raised' and focus on the resolution's substantive content, decisions, and numerical data where applicable. Omit procedural details like voting records and amendments, focusing only on the original resolution text. Ensure the output is concise yet comprehensive. Here's the resolution: {resolution}

As shown above is our resolution summarization prompt template. Its primary purpose is to condense lengthy resolution texts into a usable length while preserving their original format. As a resolution of the European Parliament, its most distinctive linguistic feature is starting with "The European Parliament", and here we require that it is immediately followed by the verb "raised". We also require it to focus on the resolution's substantive content, decisions, and numerical data where applicable. Omit procedural details like voting records and amendments, focusing only on the original resolution text. Additionally, we require it to ensure the output is concise yet comprehensive.

> **Title:**
> {title}
>
> **Resolution:**
> {resolution}
>
> **Debate:**
> {debate}
>
> **Instructions:**
> Based on the provided Title, Resolution, and Debate, compose a neutral background summary

(under 50 words) objectively describing the contextual factors that led to this issue being raised in the European Parliament. The summary must:
1. Focus solely on documented events and conditions prior to parliamentary consideration
2. State the general topic area for parliamentary discussion 3. Avoid all reference to debate content or resolution outcomes

**Output Requirements:**
- Strict 50-word maximum, in one paragraph, without title or line changes.
- First part: Factual description of pre-existing conditions (events/institutional/geopolitical context)
- Second part: Clear statement of the general discussion topic ("The Parliament will discuss...")
- Use only verified facts - no speculative language ("may reflect"/"could indicate")
- Maintain complete neutrality, exclude any reference to:
Parliamentary proceedings
Debate positions
Resolution content
Political motivations

As shown above is our background prompt template, which summarizes relevant background knowledge related to the issue based on the issue title, resolution, and full debate record, clarifying the problems the European Parliament needs to address. We require the generated background to meet the following criteria: Based on the provided Title, Resolution, and Debate, compose a neutral background summary (under 50 words) objectively describing the contextual factors that led to this issue being raised in the European Parliament. The summary must focus solely on documented events and conditions prior to parliamentary consideration, state the general topic area for parliamentary discussion, and avoid all reference to debate content or resolution outcomes. The output must adhere to a strict 50-word maximum, consist of one paragraph without title or line changes, begin with a factual description of pre-existing conditions (events/institutional/geopolitical context), and conclude with a clear statement of the general discussion topic ("The Parliament will discuss..."). Use only verified facts, no speculative language ("may reflect"/"could indicate"), while maintaining complete neutrality and excluding any reference to parliamentary proceedings, debate positions, resolution content, or political motivations.

**Topic:**
{topic}
**Resolution:**
{resolution}
**Debate:**
{debate}
**Score (0 - 9):**
{score}
**Instructions:**
{instructions}

Our opinion summarization prompt template is quite simple, and just needs to summarize each party's stances conditioned on the issue title, resolution summary, all the debate records, and the party's voting score. The key point is the instructions, which have been outlined below:

If the debate is empty or the {party} party has no arguments, output: "None"
Otherwise:
1. **Score-Specific Requirements**:

| 9-10 | Perfect alignment | None | Forbidden | "fully endorses", "perfectly aligns" |
|---|---|---|---|---|
| 7-8 | Strong support | ≤1 minor suggestion | Forbidden | "strongly supports", "approves" |
| 5-6 | General support | ≤2 constructive mods | ≤1 phrased as concern | "supports with suggestions", "advises" |
| 3-4 | Reserved approval | ≤3 major changes | ≤2 objections | "conditionally accepts", "requests revisions" |
| 0-2 | Explicit opposition | N/A | Primary focus | "rejects", "opposes fundamentally" |

2. **Argument Processing Rules**:
- For scores ≥7:
- Convert all criticism to "enhancement opportunities" (e.g., "opposes X" → "proposes strengthening X")
- Minimum 3:1 support-to-modification ratio
- For scores ≤3:
- Highlight contradictions with party principles
- Use comparative language: "fails to address", "inconsistent with"

3. **Language Enforcement**:
- **High Scores (7-10)**:
- Mandatory reinforcement phrases:
"This aligns perfectly with party's longstanding commitment to..."
"The resolution effectively advances party's priority of..."
- **Low Scores (0-3)**:
- Required framing:
"This fundamentally conflicts with party's position that..."
"The proposal overlooks critical aspects such as..."

4. **Output Validation Checklist**:
- All viewpoints begin with "{party} [score-appropriate verb]..."
- Modification proposals include concrete wording (e.g., "amend Article 3 to specify...")
- For scores ≥5, ≥80% of content must directly affirm resolution goals
- Opposition arguments (if allowed) must reference specific resolution clauses

5. **Examples (Score=8)**:
- {party} strongly supports the transparency measures in Articles 1-3, particularly the lobbyist disclosure requirements.
- {party} proposes extending document publication deadlines by 15 days to ensure thorough review without opposing the principle.
- {party} applauds the anti-fraud provisions as matching their 2023 manifesto commitments.

6. **Special Cases**:
- Empty debate with score ≥7 → "Consistent silent endorsement"
- Contradictory arguments → Flag with: "[Note: Reconcile with score {score}]"
7. Output format (one viewpoint per line):
viewpoint_1
viewpoint_2
...
(Max 5 viewpoints, no numbering or bullets)

The instructions outlined above illustrate how to determine different parties' stances based on parliamentary debate records. If the debate lacks content or the party has no arguments, the output is "None". Otherwise, the system categorizes responses into specific score ranges with criteria for alignment, support, and opposition. High scores (9-10) indicate perfect alignment, while low scores (0-2) reflect explicit opposition. The prompt includes rules for processing arguments, emphasizing positive reinforcement for higher scores and highlighting contradictions for lower scores. An output validation checklist ensures all viewpoints are appropriately framed and modifications are clearly articulated. Examples illustrate these rules, and special cases address unique situations. The output format presents up to five viewpoints per line for clarity and coherence.

## C.2 TASK PROMPTS

In this section, we will present the prompts we used in the process of political consensus finding with LLMs using *PoliCon*.

> You are an AI designed to provide the most helpful, clear, and concise responses. Focus on giving actionable information, ensuring accuracy and detail without overwhelming the user. You should also be patient, polite, and calm. Avoid unnecessary complexity and always prioritize practical, user-friendly advice.

The system prompt displayed above demonstrates good performance and has been widely used in previous work (Zhang et al., 2025).

In the following, we will present our task prompt. We require the creation of a consensus European Parliament resolution statement that adheres to the specified criteria in a single, unbroken paragraph. The statement must begin with "The European Parliament raised" and concentrate on the substantive content, decisions, and numerical data where applicable. It should address opposing stances by providing detailed solutions and mitigations for the concerns raised, while moderating supporting stances with appropriate qualifications and limitations. Procedural details such as voting records and amendments should be omitted, focusing solely on the original resolution text. The output must be concise yet comprehensive.

> Background: {background}
> A group of {party_num} political parties in the European Parliament was required to find consensus on this topic: {topic}.
> Below is each party's stance:
> {stances}
> {task_requirements}
> Your task is to write a consensus European Parliament resolution statement that meets the upper requirements in one continuous paragraph, without any formatting or line breaks. Begin the resolution statement with 'The European Parliament raised' and focus on the resolution's substantive content, decisions, and numerical data where applicable. **It is forbidden to include the name of any party.** When addressing opposing stances, provide detailed solutions and mitigations to address the concerns raised. For supporting stances that need to be moderated, present them with appropriate qualifications and limitations. Omit procedural details like voting records and amendments, focusing only on the original resolution text. You don't need to make analysis about partys' stances but just need to write a resolution statement. Ensure the output is concise yet comprehensive. Here's an example of the resolution:
> {resolution}
> Now is your turn:

As for our opinion prompt template, we just simply use the following format to illustrate each party's positions:

> Party {party_name}: {stance}

### C.2.1 TASK REQUIREMENT TEMPLATES

In this section, we will present our prompts for conveying the specific task requirements to the tested LLMs.

First, we introduce our task description for different political goals. The first is the prompt for passing the resolution under specific voting requirements:

> The resolution should be based on the seat proportions of each party, where the proportions are defined by the following weights rather than actual parliamentary seats: {sea_apportionmen_weights}

Then are the requirements for Rawlsianism and Utilitarianism:

> The resolution should be based on the Rawlsian principle, which means maximizing the benefits for the political parties with the weakest positions to achieve the greatest possible fairness.

> The resolution should be based on the Utilitarianism principle, which means maximizing the preferences of all political parties while ensuring their sum is maximized.

Finally, we convey each party's seat apportionment information through the following simple template:

> Party {party_name}'s seat proportion is {seat_proportion}%.

### C.2.2 Voting Mechanism Templates

In this section, we introduce the prompts used in the paper to describe three different voting mechanisms. Below are the details for simple majority, two-thirds majority, and veto power, respectively:

> The resolution should be supported by more than 50% of the parliament members. The resolution should reflect each party's stance in proportion to their seat allocation.

> The resolution should be supported by more than two-thirds of the parliament members. The resolution should reflect each party's stance in proportion to their seat allocation.

> Because {veto_party_name} has veto power, the resolution should meet the stances of {veto_party_name} as much as possible while also being supported by more than 50% of the parliament members. The resolution should reflect each party's stance in proportion to their seat allocation.

### C.3 Evaluation Prompts

In this section, we will present our open-ended evaluation framework's prompts.

> Your task is to give a score for the following statement based on the score guidelines and your knowledge of the party.
> {{"{wait_for_judge_statement}"}}.
> {{"choices": ["not at all", "very poorly", "poorly", "somewhat poorly", "somewhat", "moderately", "somewhat well", "well", "very well", "perfectly"], "choice_numbers": [0, 1, 2, 3, 4, 5, 6, 7, 8, 9]}}.
> Your score should in [0, 1, 2, 3, 4, 5, 6, 7, 8, 9], and explain why you give this score in one sentence. The reason should be specific to the party's stance.
> Please respond in the following JSON format:
> {{"explanation": "<one sentence explanation>","score": <choice_number>}}

> Your task is to give a score for the following statement based on the score guidelines and your knowledge of the party.
> {{"{feasibility_judge_statement}"}}.
> {{"choices": ["not at all", "very poorly", "poorly", "somewhat poorly", "somewhat", "moderately", "somewhat well", "well", "very well", "perfectly"], "choice_numbers": [0, 1, 2, 3, 4, 5, 6, 7, 8, 9]}}.
> Your score should in [0, 1, 2, 3, 4, 5, 6, 7, 8, 9], and explain why you give this score in one sentence. The reason should be specific to the party's stance and the background information.
> Please respond in the following JSON format:
> {{"explanation": "<one sentence explanation>","score": <choice_number>}}

The templates presented above are resolution-stance alignment evaluation and resolution feasibility evaluation in our comprehensive evaluation framework, focusing primarily on the content awaiting assessment and the instructions for evaluation. The subsequent prompt provides a structured approach for scoring statements based on specific guidelines and party knowledge. It includes a scoring system with choices ranging from "not at all" to "perfectly", corresponding to numerical values from 0 to 9. The task requires a precise response within this numerical range, ensuring alignment with the party's principles and facilitating consistent evaluation outcomes. The most crucial parts of the template are the content awaiting evaluation and the evaluation guidelines, which we will present separately below.

The first is the template we provide to our evaluation framework for assessing content. This framework is designed to analyze the degree to which a given resolution encapsulates the specific implementation details that reflect the stances of a particular party, regardless of whether these stances support or oppose the issue, and verify the feasibility of the resolution based on the given background and the party's stance. By evaluating the alignment and practical feasibility of the resolution in relation to the party's opinion and the broader contextual background, the framework seeks to provide a comprehensive assessment of the extent to which the resolution embodies and advances the party's core principles and strategic priorities. The guidelines included in the prompt serve to direct the assessment process, ensuring consistency and accuracy in evaluating the alignment between the resolution and the party's stance.

> Background:
> {{Begin of the background}}
> {background}
> {{End of the background}}
> Consider the following statement:
> {{Begin of the resolution}}
> {resolution}
> {{End of the resolution}}
> The {party_name}'s opinion is:
> {{Begin of the stance}}
> {stance}
> {{End of the stance}}
> To what extent does this resolution provide specific implementation details that capture {party_name}'s stances? **Regardless of whether the stances itself is supportive or opposing to the issue.**
> {guidelines}

> Background:
> {{Begin of the background}}
> {background}
> {{End of the background}}
> Consider the following statement:
> {{Begin of the resolution}}
> {resolution}

{{End of the resolution}}
The {party_name}'s opinion is:
{{Begin of the stance}}
{stance}
{{End of the stance}}
To what extent is this resolution feasible based on the background and party_name's stance?
**You don't need to evaluate if the resolution aligns with the stance, you should only evaluate the feasibility of the resolution itself.**
{guidelines}

The next one is our evaluation guidelines, which aim to assess resolutions based on their alignment with the European Parliament's stances and feasibility with the given background. The scoring system ranges from 0 to 9, evaluating resolutions on their specificity, feasibility, and comprehensiveness in addressing key points from various stances. Scores from 0-3 indicate resolutions that lack proper format, omit critical details, or are largely disconnected from the background and stance. Scores from 4-6 reflect partial alignment with party interests, addressing some but not all key aspects. Or the overall direction is correct, but insufficient details reduce practical viability. Scores from 7-9 recognize fully detailed and practical implementation measures that comprehensively address all stance points, ensuring no compromise or dilution of objectives and offering realistic and implementable measures with a clear pathway.

Please follow this scoring guideline:
- **Score 0-3**: If the resolution does not start with "The European Parliament", or if the resolution only rephrases content from the stances without providing specific implementation details, contains impractical/unfeasible implementation proposals, omits key points mentioned in the stances, or if it contains elements that weaken/dilute the benefits sought in supportive stances (for opposing stances, if it promotes/strengthens what the party opposes), or it doesn't match the topic about the stances. IF THE CONTENT IS EVEN NOT IN A RESOLUTION FORMAT, YOU SHOULD GIVE 0 DIRECTLY.
- **Score 4-6**: If the resolution provides some feasible implementation details for the stances' requirements but lacks comprehensiveness (e.g. only addresses some aspects, missing some points from the stances) or contains minor conflicts with party interests (e.g. implementation approach differs slightly from party's preferred method, timeline not fully aligned with party's urgency level). The resolution should cover at least half of the key points mentioned in the stances.
- **Score 7-9**: If the resolution provides detailed, concrete and practically feasible implementation measures that fully strengthen and implement supportive stances (for opposing stances, score high if the resolution effectively addresses and resolves the opposition's concerns) without any dilution or compromise. The resolution must comprehensively address ALL points raised in the stances, with higher scores for more detailed coverage of each point.

Please follow this scoring guideline:
- **Score 0-3**: If the resolution is largely disconnected from the background and stance, proposes clearly unfeasible measures, or if there are contradictions/inconsistencies in the resolution, or if it ignores practical constraints (resources, timeframe, policy frameworks, international context), provides only idealistic or unrealistic goals without actionable pathways. IF THE CONTENT IS EVEN NOT IN A RESOLUTION FORMAT, YOU SHOULD GIVE 0 DIRECTLY.
- **Score 4-6**: If the resolution shows partial consideration of the background and stance but lacks comprehensiveness. The measures are somewhat feasible but the implementation pathway is vague or insufficiently adapted to real-world conditions. It contains minor inconsistencies with the background. The overall direction is correct, but insufficient details reduce practical viability.
- **Score 7-9**: If the resolution fully integrates the background conditions and stance requirements, offering realistic and implementable measures with a clear pathway. For sup-

> portive stances, the resolution effectively advances the goals within the existing background. For opposing stances, it addresses and mitigates the concerns raised in the stance, ensuring alignment between the background and party position. It comprehensively covers all key points from the stance, provides solutions to background-related constraints, and demonstrates precise, strong, and highly feasible execution plans.

In our implementation, we use the average of the two metrics as the final score. The experiments in subsection 4.1 and Appendix F.1 demonstrate a strong consistency between our evaluation method and the real voting results.

## D    QUALITY OF THE CLEANED DATA

In this section, we will present specific cases to demonstrate that our data cleaning process did not introduce noticeable biases from using LLMs, thereby providing a more intuitive validation of the cleaned data quality.

> **Original debates:**
>
> *Kristiina Ojuland (ALDE):* Mr President, the financial discipline proposal was created to ensure that obligations in regard to direct payments for every year are fully respected. Unfortunately, the Commission has gone further and is now asking for the market crisis reserve to be composed from the CAP budget. The near 5 % reduction in direct payments to farmers receiving more than EUR 5 000 would give an overall reduction of nearly EUR 1.5 billion. That is almost double the amount that the Commission was looking to cover initially. I understand that it is looking to cut as much as possible from every budget line, but this time the Commission has gone too far. I would like to call on colleagues to support the rapporteur and the Commission's initial proposal to cut the payments by 0.7% and include only those farmers receiving more than EUR 5 000. I do not disagree with the market crisis reserve, but it should be composed from different funds.
>
> *James Nicholson (ECR):* With the threshold of EUR 5000 proposed in the report, 80% of people would be excluded from financial discipline. Now I do not know about you, but I would find it extremely difficult to justify to any of the farmers that I represent why 20 % of them will be required to cover the deficit which is incurred because of spending which could be attributed to 100 % of the Member States and 100 % of the farmers. We should not support a measure which will undoubtedly lead to such extreme inequalities between farmers and between Member States, and it certainly destroys any hope of a level playing field. On the Direct Payments Regulation which is currently part of the CAP reform dossier, this Parliament voted to delete the financial discipline threshold. The objective of this mandate was to ensure that the burden of financial discipline is shared equally and fairly between farmers across the European Union.

> **Extracted stances:**
>
> *ALDE:* ALDE suggests limiting direct payment reductions to the Commission's initial 0.7% threshold for farmers above 5000, opposing the higher 5% cut to prevent disproportionate budgetary impacts on agricultural stakeholders.
>
> *ECR:* ECR rejects the proposed EUR 5000 threshold as firmly doubting with their principle of equal burden-sharing across all agricultural producers.

The original debates and corresponding stances extracted by LLMs shown above are extracted from agricultural topic data. A comparative analysis shows that the LLMs-extracted stances accurately capture the core positions articulated in the original debates. Importantly, the model did not introduce noticeable biases of its own during the extraction process, which indicates its reliability in capturing

the argumentative content while maintaining fidelity to the original discourse. Therefore, the extracted data by LLMs can serve as a trustworthy and precise foundation for subsequent evaluation and analysis.

To further verify the impact of DeepSeek-R1 on our data-cleaning pipeline, we conduct a human evaluation experiment to measure the subjective quality of the data cleaned by DeepSeek-R1. We recruit ten human annotators and design an intuitive and easy-to-use interface for the experiment: the interface displays the original and cleaned resolutions and stances, and annotators are asked to evaluate the quality of the LLM-based data cleaning for each sample using a scalar score from 0 to 5. We randomly sample 100 instances from the dataset and assign 10 instances to each annotator as their labeling task. The results show that human annotators rate more than 95.8% of the cleaned resolution samples and more than 75% of the cleaned stances samples at 3 points or above. This indicates that DeepSeek-R1 is highly effective in improving data-cleaning quality and achieves an acceptable level of subjective quality. Details of the interface design appear in Appendix E.2.

## E  EXPERIMENTAL DETAILS

In this section, we will introduce more details about how we implement our experiments.

### E.1  DETAILED SIMULATED EVALUATION CONSISTENCY CALCULATION PROCESS

We primarily used `gaussian_kde` from `scipy.stats` for error computation. Specifically, the complete computation process can be divided into the following three steps: (1) compute the errors; (2) compute the standard deviation of the errors; (3) apply Gaussian kernel density estimation (Gaussian KDE) to obtain a smooth estimate of the error distribution, which is then used for visualization. The detailed explanation of the formulas and procedures for each step is as follows:

**Error Definition and Computation.**  We define the error as the difference between the predicted score of our judge model and the real-world ground truth. The strict definition is as follows: suppose there are $n$ observed samples, and denote the set of sample indices as $\mathcal{I} = 1, 2, \ldots, n$. For each $i \in \mathcal{I}$, let $p_i \in [0, 9] \cap \mathbb{N}$ be the predicted score, and $g_i \in [0, 9] \cap \mathbb{N}$ be the corresponding ground-truth score, where $\mathbb{N}$ is the set of natural numbers. Then the prediction error of the $i$-th sample is defined as: $e_i = p_i - g_i$.

**Standard Deviation of the Error.**  We first compute the estimation of the errors $\bar{e} = \frac{1}{n} \sum_{i=1}^{n} e_i$, and then compute its standard deviation $\sigma = \sqrt{\frac{1}{n} \sum_{i=1}^{n} (e_i - \bar{e})^2}$ with the estimation.

**Using Gaussian KDE to Obtain and Visualize the Probability Density Function.**  To obtain a smooth estimate of the error distribution, we apply Gaussian KDE to the empirical sample $\{e_i\}_{i=1}^{n}$, which is a method that treats each sample point as the center of a Gaussian distribution and forms a mixture of Gaussians to estimate the probability density function of a random variable. It is typically defined by the following formula:

$$\hat{f}_h(x) = \frac{1}{nh\sqrt{2\pi}} \sum_{i=1}^{n} \exp\left( -\frac{1}{2} \left( \frac{x - e_i}{h} \right)^2 \right),  \tag{2}$$

where $h > 0$ is the bandwidth (smoothing) parameter. In this paper, we implement this function using `gaussian_kde` from `scipy.stats`, where $h$ is automatically determined by the Scott method (Scott, 2015) within `gaussian_kde`. Finally, we set $x \in [-9, 9]$ with interval = 0.001 for sampling and computation, thereby obtaining the error probability density curve shown in Figure 4.

### E.2  DETAILS FOR OUR HUMAN STUDY

Our human study can be divided into two main parts. The first part investigates whether the evaluations produced by our evaluator are consistent with the ratings given by human annotators. The second part examines the quality of the data cleaned by DeepSeek-R1. In the following sections, we provide a detailed description of our annotation pipeline and the user interface used in the study.

**Does Our Evaluator Align with Human Annotators?**    To verify whether our proposed evaluator can reliably score the consensus resolutions generated by LLMs, we design a concise and intuitive user interface that allows annotators to manually assess the degree to which model-generated resolution texts satisfy the party stances. As shown in Figure 11, the interface includes the following key components:

- **Problem Background and Task Information Display.** The top of the interface shows the title of the current topic, relevant background information, and the participating political parties along with a summary of their positions, ensuring that annotators can make quality judgments based on a full understanding of the context.

- **Model-Generated Resolution Display.** Annotators can view the full resolution text generated by the LLM and assess its consistency with the positions of each party, as well as the feasibility and reasonableness of the resolution content.

- **Scoring Area.** To improve the reproducibility of scoring, we provide clear explanations of the score ranges (e.g., 0 to 9 corresponding to different levels of consistency) and require annotators to give separate scores for two dimensions: alignment, i.e., the extent to which the resolution accurately reflects and integrates the positions of various parties; and feasibility, i.e., whether the resolution is realistically implementable and logically coherent.

- **Multilingual Support.** Considering that annotators may come from different native language backgrounds, we added a multilingual mode switch to the interface to improve usability and robustness of the annotation process.

Since our task involves the intersection of AI and political science, we recruited 20 human annotators, half of whom have an AI-related background and have obtained or are pursuing a PhD; the other half are law students, enabling coverage of a broader range of relevant domain knowledge.

We randomly selected 100 consensus resolutions generated by LLMs, and to ensure robustness of the evaluation, each sample was repeated three times and assigned to three different annotators. Using this UI, we obtained a total of 300 human annotations. Statistical results show that the average error between our evaluator and human annotators is only 1.61, with over 72% of sample errors falling within the $\pm 1.92$ range, indicating that the evaluator maintains good consistency with human judgments.

**Quality of DeepSeek-R1 Data Cleaning.**    To evaluate whether the structured content generated by DeepSeek-R1 during the data cleaning stage is sufficiently reliable, we further conducted a human quality assessment experiment. As shown in Figure 12, we designed a dedicated user interface for rating the quality of the cleaning results, which mainly consists of the following components:

- **Original and Cleaned Text Comparison Area.** The interface clearly displays three types of information side by side: the original parliamentary resolution, the original party stances, and the cleaned data version generated by DeepSeek-R1. This side-by-side layout allows annotators to quickly judge whether the cleaning results are faithful, accurate, and free of missing important information.

- **Single-Dimension Quality Scoring Area.** Annotators are required to assign an overall score from 0 to 5 for the cleaning quality of each sample, with explanations provided for the scores (e.g., 0 for severe errors, 5 for fully accurate).

Naturally, this UI also supports multilingual functionality, enabling displays in the native languages of different annotators. We recruited ten annotators and randomly selected 100 samples, assigning 10 samples to each annotator via the UI for independent annotation. The final experimental results indicate that human annotators rated over 95.8% of the resolution-cleaned samples and over 75% of the stance-cleaned samples with a score of 3 or higher. These findings demonstrate that DeepSeek-R1 is highly effective in ensuring the quality of cleaned data, achieving an acceptable, and in many cases, a favorable level in subjective quality assessments.

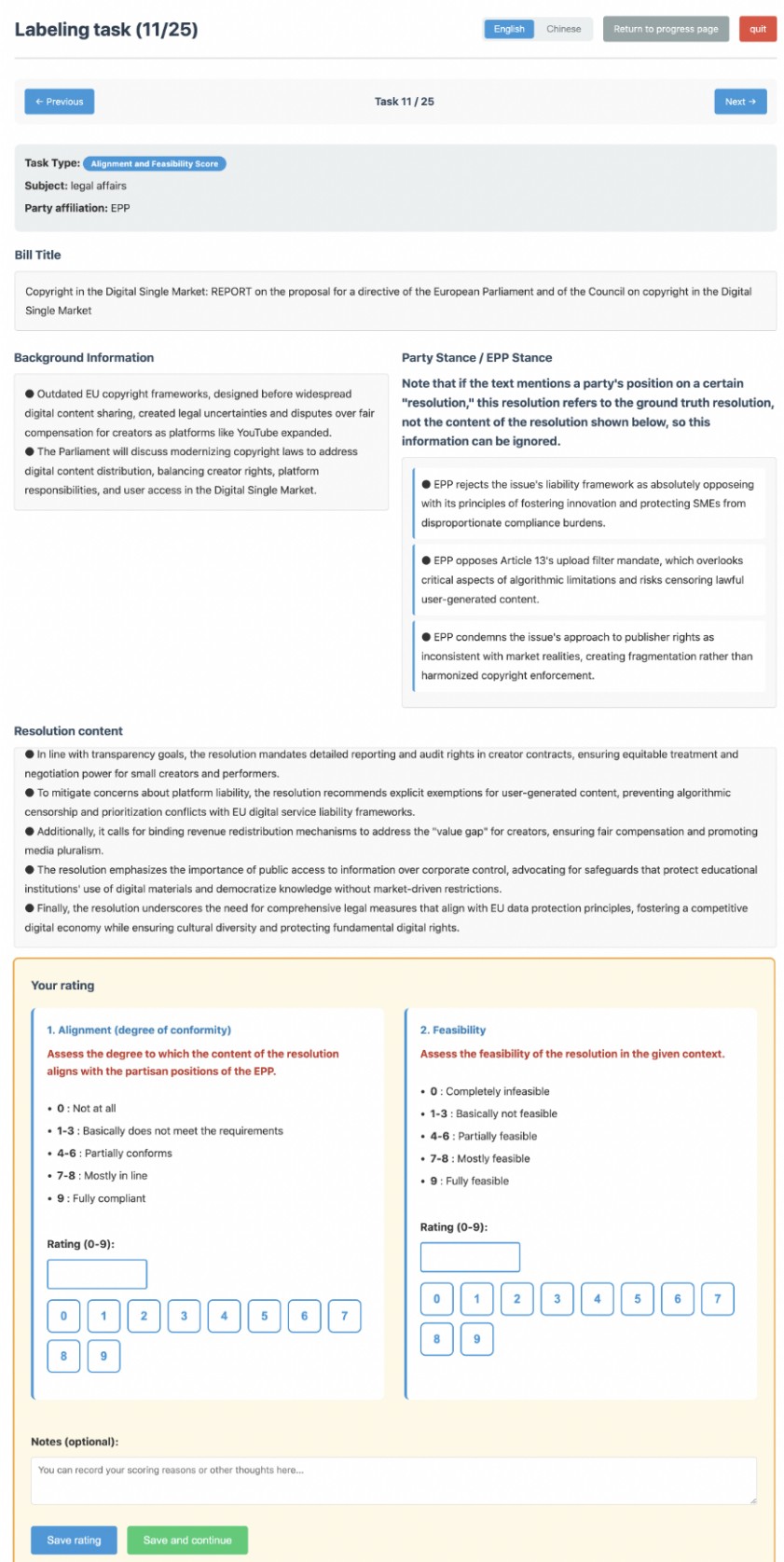

Figure 11: The UI for the human evaluation for our evaluator.

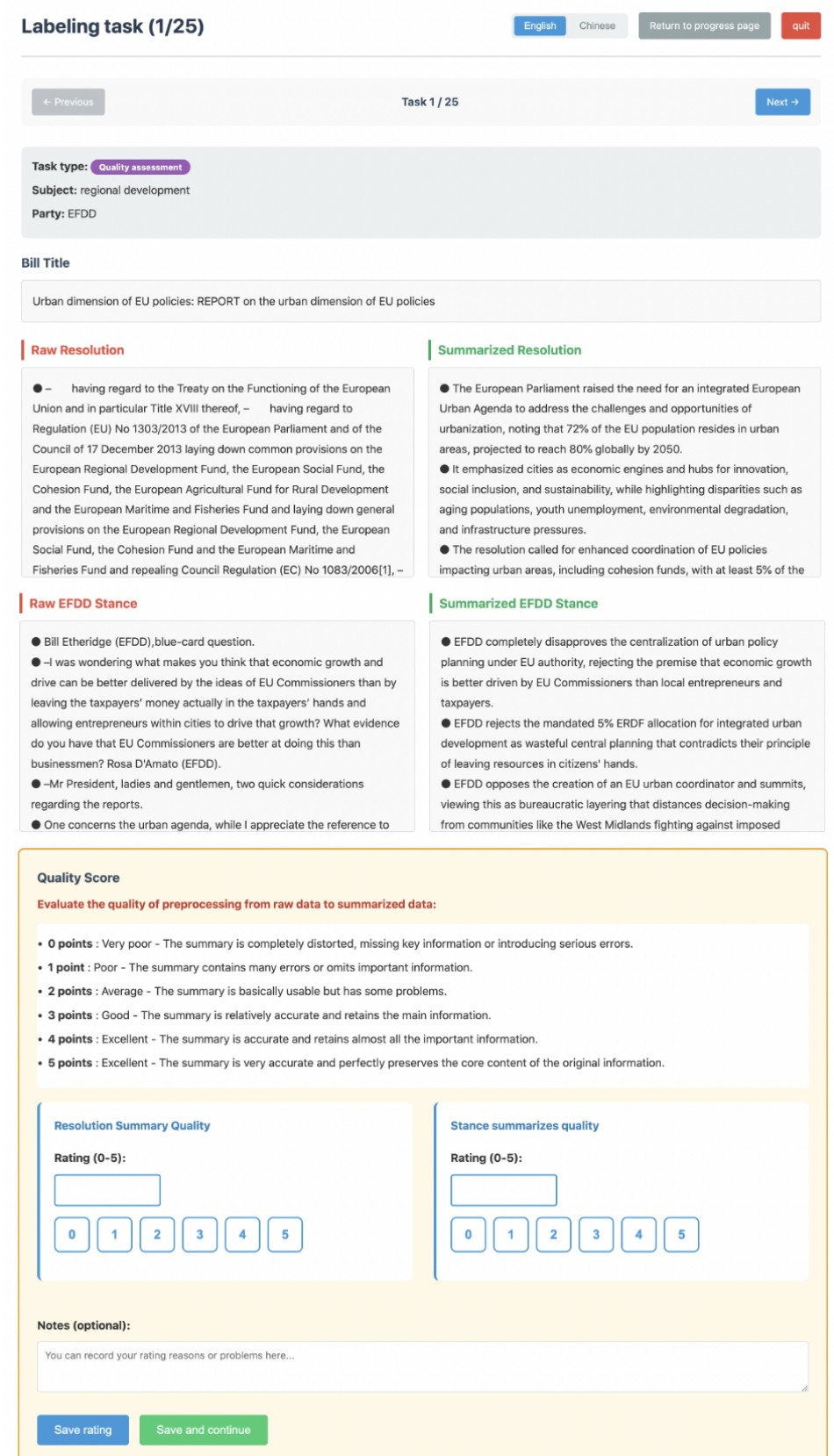

Figure 12: The UI for the data cleaning quality evaluation.

## F    MORE EXPERIMENTAL RESULTS

In this section, we will illustrate more experimental results, especially more simulated consistency results of our open-ended evaluation framework and detailed performance on all the fine-grained topics.

### F.1    DETAILED SIMULATED EVALUATION CONSISTENCY RESULTS

In this section, we will provide a more detailed presentation and supplement to the experimental results from subsection 4.1.

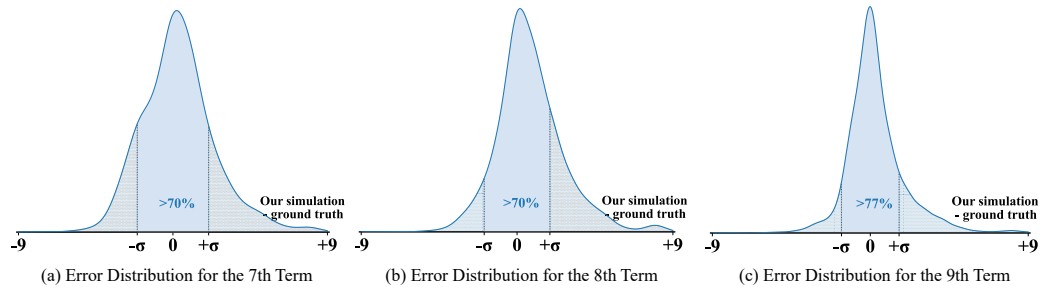

(a) Error Distribution for the 7th Term    (b) Error Distribution for the 8th Term    (c) Error Distribution for the 9th Term

Figure 13: Error distribution on our simulated votes and the ground truth for the 7th (2009 - 2014), 8th (2014 - 2019), and half of the 9th (2019 - 2022) parliament terms.

As shown in Figure 13, we employed the same method as in subsection 4.1 to further illustrate the consistency between the simulated voting results of our open-ended evaluation framework and the actual voting results for the 7th, 8th, and 9th terms of the European Parliament.

The error is calculated by subtracting the ground truth voting score from the simulated score of our sub-evaluation module. It can be observed that for each term of the parliament, most of our simulation results fall within the ground truth's $\sigma$ range: 70% for the 7th, 70% for the 8th, and 77% for the 9th. From this more detailed error analysis, we can also see that in almost every parliament, our evaluator tends to overestimate slightly rather than underestimate, particularly evident in the 8th parliament. This may be related to factors such as the sycophancy of LLMs (Sharma et al., 2023; Kran et al., 2025).

### F.2    IMPACT OF PARTY SEATS ON VETO POWER

To examine how the number of seats held by each party affects the robustness of the task of veto power, we design two experiments. The first tests whether an increasing sample number in each party's seats strengthens or weakens the robustness of veto power; the second tests how strongly different party seat allocations influence the robustness of the final outcome.

For the first experiment, since we randomly draw one seat allocation for each data point, computing each model's overall expected outcome over a large set of data with varying seat distributions is clearly a robust way to test the model's veto power. Therefore, if in an experiment where we sample multiple different numbers of seats for each issue and compute the corresponding expected outcomes, the resulting experimental findings are consistent with the performance of the models presented in Table 1, this would indicate that increasing the number of party seats can enhance the robustness of the veto power outcomes.

In one of our task scenarios, namely the development subtopic with 4 parties, we randomly generated 30 different party seat allocations for each data point and conducted experimental analysis on the veto power objective.

As shown in Table 10, we present the mean and variance for different seat allocation outcomes. By examining the experimental results in the table, we find that they are entirely consistent with those reported in Table 1: Qwen2.5-32B, Llama-3.3-70B, and Qwen2.5-72B exhibit similar performance,

Table 10: The VP score with 30 samples for each issue on the 'development' topic.

| Model | VP Score |
|---|---|
| Qwen2.5-32B | $0.53 \pm 0.04$ |
| Llama-3.3-70B | $0.52 \pm 0.04$ |
| Qwen2.5-72B | $0.53 \pm 0.03$ |
| GPT-4o | $0.56 \pm 0.04$ |
| Deepseek-V3.1 | $0.57 \pm 0.04$ |
| Gemini-2.5 | $0.59 \pm 0.03$ |

which is lower than that of the three commercial models. For these three commercial models, this capability, from strongest to weakest, is: Gemini-2.5, Deepseek-V3.1, and GPT-4o. This finding corroborates our first experimental conclusion, namely that increasing the number of seats sample number held by a party will enhance, rather than diminish, the robustness of the veto power task.

For the second experiment, we can conduct the analysis by examining the variances in the table above. It can be seen that all the variances are kept within a controllable range. This result demonstrates that although different choices for the number of party seats do indeed affect the outcomes of the veto power task to some extent, the impact is not significant and remains within a relatively controllable range.

Combining the two experiments above, we arrive at the following two conclusions: (1) sampling different numbers of seats for each party can significantly enhance the robustness of the experimental results on veto power; and (2) the choice of specific party seat allocations has only a minor impact on the final results.

### F.3 DETAILED FINE-GRAINED TOPICS RESULTS

In this section, we will demonstrate the performance of different LLMs on each fine-grained topic, as shown in Table 11 to Table 15.

As shown in Table 11, in the economic topics of the euro, including international trade, internal market & consumer protection, employment & social affairs, and economic & monetary affairs (with some topic names abbreviated for convenience in the table), there are significant performance differences among various LLMs. Overall, Gemini-2.5 performs best among all models, especially in SM and Rawls tasks, achieving the highest score of 0.86-0.90 and 3.11-5.04. DeepSeek-V3.1 and GPT-4o follow closely, performing strongly across all five domains. Notably, GPT-4o shows robustness in the Util task, while DeepSeek-V3.1 maintains high stability across domains. Meanwhile, as task difficulty increases (such as with the 2/3M and Veto tasks), the performance of all models declines significantly. For instance, the pass rate of Qwen2.5-32B in the 2/3M task drops to 0.32-0.40 compared to the SM task. Additionally, the type of topic significantly impacts performance, with policy-related topics like employment & social affairs and economic & monetary affairs generally being more challenging than industry development topics like international trade and internal market & consumer protection, highlighting the challenges LLMs face with complex political issues.

Table 11: Performance of different LLMs on *PoliCon*'s Economic topic. The values in square brackets indicate the range of each metric, and all metrics follow the principle that higher values are better. The background color of the table cells deepens as the performance improves. The blue color scheme represents metrics in the 0-1 range, while the red color scheme represents metrics in the 0-9 range.

| Topic | Model | SM [0-1] ↑ | | | 2/3M [0-1] ↑ | | | VP [0-1] ↑ | | | Rawls [0-9] ↑ | | | Util [0-9] ↑ | | |
|---|---|---|---|---|---|---|---|---|---|---|---|---|---|---|---|---|
| | | 2 | 4 | 6 | 2 | 4 | 6 | 2 | 4 | 6 | 2 | 4 | 6 | 2 | 4 | 6 |
| international trade | Qwen2.5-32B | 0.79 | 0.81 | 0.82 | 0.28 | 0.36 | 0.26 | 0.60 | 0.56 | 0.59 | 4.29 | 3.58 | 2.89 | 6.18 | 6.20 | 6.03 |
| | Llama-3.3-70B | 0.76 | 0.77 | 0.80 | 0.37 | 0.39 | 0.31 | 0.59 | 0.57 | 0.57 | 4.15 | 3.70 | 2.98 | 6.24 | 6.45 | 6.26 |
| | Qwen2.5-72B | 0.79 | 0.79 | 0.87 | 0.31 | 0.41 | 0.26 | 0.63 | 0.57 | 0.59 | 4.34 | 3.57 | 3.03 | 6.08 | 6.19 | 6.04 |
| | GPT-4o | 0.88 | 0.88 | 0.95 | 0.63 | 0.55 | 0.44 | 0.68 | 0.65 | 0.77 | 5.07 | 4.47 | 3.51 | 6.70 | 6.69 | 6.50 |
| | DeepSeek-V3.1 | 0.95 | 0.89 | 0.92 | 0.66 | 0.57 | 0.49 | 0.72 | 0.68 | 0.74 | 4.90 | 4.21 | 3.30 | 6.57 | 6.67 | 6.60 |
| | Gemini-2.5 | 0.97 | 0.96 | 0.90 | 0.70 | 0.65 | 0.41 | 0.84 | 0.62 | 0.75 | 5.60 | 4.69 | 3.64 | 6.70 | 6.64 | 6.30 |
| internal market | Qwen2.5-32B | 0.84 | 0.82 | 0.93 | 0.60 | 0.61 | 0.50 | 0.65 | 0.74 | 0.61 | 5.40 | 4.52 | 3.64 | 6.94 | 7.02 | 6.79 |
| | Llama-3.3-70B | 0.82 | 0.84 | 0.95 | 0.69 | 0.62 | 0.61 | 0.66 | 0.74 | 0.64 | 5.59 | 4.66 | 3.84 | 7.04 | 7.23 | 7.01 |
| | Qwen2.5-72B | 0.84 | 0.85 | 0.95 | 0.68 | 0.61 | 0.57 | 0.65 | 0.75 | 0.66 | 5.42 | 4.69 | 3.80 | 6.86 | 7.10 | 6.94 |
| | GPT-4o | 0.86 | 0.87 | 0.95 | 0.75 | 0.72 | 0.73 | 0.72 | 0.79 | 0.73 | 5.83 | 4.89 | 4.05 | 7.21 | 7.30 | 7.19 |
| | DeepSeek-V3.1 | 0.91 | 0.90 | 0.98 | 0.75 | 0.75 | 0.73 | 0.71 | 0.77 | 0.77 | 5.88 | 5.00 | 4.07 | 7.21 | 7.37 | 7.13 |
| | Gemini-2.5 | 0.94 | 0.98 | 0.98 | 0.79 | 0.77 | 0.68 | 0.62 | 0.79 | 0.66 | 6.34 | 5.16 | 4.32 | 7.31 | 7.32 | 7.09 |
| employment | Qwen2.5-32B | 0.66 | 0.67 | 0.90 | 0.41 | 0.44 | 0.44 | 0.46 | 0.49 | 0.62 | 3.61 | 2.84 | 2.26 | 6.11 | 6.30 | 6.52 |
| | Llama-3.3-70B | 0.67 | 0.67 | 0.92 | 0.43 | 0.42 | 0.64 | 0.49 | 0.51 | 0.64 | 3.82 | 2.76 | 2.38 | 6.24 | 6.40 | 6.64 |
| | Qwen2.5-72B | 0.80 | 0.76 | 0.95 | 0.48 | 0.55 | 0.69 | 0.61 | 0.56 | 0.62 | 4.51 | 3.40 | 2.28 | 6.52 | 6.69 | 6.51 |
| | GPT-4o | 0.70 | 0.75 | 0.95 | 0.51 | 0.47 | 0.64 | 0.61 | 0.51 | 0.62 | 4.23 | 2.78 | 2.38 | 6.52 | 6.50 | 6.68 |
| | DeepSeek-V3.1 | 0.79 | 0.84 | 0.92 | 0.48 | 0.47 | 0.64 | 0.59 | 0.51 | 0.62 | 4.33 | 3.35 | 2.69 | 6.62 | 6.53 | 6.72 |
| | Gemini-2.5 | 0.90 | 0.84 | 0.82 | 0.51 | 0.51 | 0.51 | 0.59 | 0.56 | 0.54 | 4.41 | 3.18 | 2.49 | 6.38 | 6.40 | 6.71 |
| economic affairs | Qwen2.5-32B | 0.73 | 0.73 | 0.73 | 0.29 | 0.32 | 0.24 | 0.45 | 0.42 | 0.51 | 3.56 | 2.86 | 2.45 | 5.87 | 6.03 | 5.99 |
| | Llama-3.3-70B | 0.70 | 0.72 | 0.76 | 0.31 | 0.42 | 0.31 | 0.42 | 0.42 | 0.53 | 3.57 | 2.83 | 2.45 | 6.02 | 6.17 | 6.20 |
| | Qwen2.5-72B | 0.74 | 0.72 | 0.75 | 0.36 | 0.41 | 0.28 | 0.42 | 0.42 | 0.52 | 3.61 | 2.80 | 2.36 | 5.87 | 6.06 | 6.00 |
| | GPT-4o | 0.80 | 0.80 | 0.80 | 0.43 | 0.48 | 0.45 | 0.51 | 0.48 | 0.55 | 3.84 | 2.94 | 2.45 | 6.15 | 6.30 | 6.33 |
| | DeepSeek-V3.1 | 0.80 | 0.86 | 0.80 | 0.41 | 0.50 | 0.42 | 0.50 | 0.50 | 0.54 | 4.07 | 2.94 | 2.57 | 6.10 | 6.29 | 6.30 |
| | Gemini-2.5 | 0.84 | 0.89 | 0.80 | 0.48 | 0.48 | 0.35 | 0.60 | 0.51 | 0.57 | 4.21 | 2.97 | 2.55 | 6.15 | 6.22 | 6.17 |
| Average | Qwen2.5-32B | 0.76 | 0.76 | 0.81 | 0.37 | 0.40 | 0.32 | 0.53 | 0.53 | 0.56 | 4.13 | 3.34 | 2.73 | 6.20 | 6.29 | 6.22 |
| | Llama-3.3-70B | 0.74 | 0.75 | 0.83 | 0.42 | 0.45 | 0.41 | 0.53 | 0.53 | 0.57 | 4.17 | 3.37 | 2.81 | 6.32 | 6.47 | 6.42 |
| | Qwen2.5-72B | 0.78 | 0.77 | 0.84 | 0.43 | 0.46 | 0.39 | 0.55 | 0.54 | 0.57 | 4.30 | 3.43 | 2.76 | 6.22 | 6.37 | 6.25 |
| | GPT-4o | 0.82 | 0.82 | **0.89** | 0.56 | 0.54 | **0.52** | 0.61 | 0.58 | **0.65** | 4.62 | 3.67 | 2.97 | **6.57** | 6.61 | 6.57 |
| | DeepSeek-V3.1 | 0.86 | 0.87 | 0.88 | 0.55 | 0.56 | **0.52** | 0.62 | **0.60** | 0.64 | 4.69 | 3.71 | 3.02 | 6.53 | **6.62** | **6.58** |
| | Gemini-2.5 | **0.90** | **0.92** | 0.86 | **0.60** | **0.58** | 0.45 | **0.67** | **0.60** | 0.63 | **5.04** | **3.85** | **3.11** | 6.56 | 6.55 | 6.44 |

Table 12: Performance of different LLMs on *PoliCon*'s Industry topic. The values in square brackets indicate the range of each metric, and all metrics follow the principle that higher values are better. The background color of the table cells deepens as the performance improves. The blue color scheme represents metrics in the 0-1 range, while the red color scheme represents metrics in the 0-9 range.

| Topic | Model | SM [0-1] ↑ | | | 2/3M [0-1] ↑ | | | VP [0-1] ↑ | | | Rawls [0-9] ↑ | | | Util [0-9] ↑ | | |
|---|---|---|---|---|---|---|---|---|---|---|---|---|---|---|---|---|
| | | 2 | 4 | 6 | 2 | 4 | 6 | 2 | 4 | 6 | 2 | 4 | 6 | 2 | 4 | 6 |
| agriculture | Qwen2.5-32B | 0.74 | 0.80 | 0.89 | 0.30 | 0.47 | 0.44 | 0.52 | 0.65 | 0.70 | 4.74 | 4.33 | 3.70 | 6.48 | 6.50 | 6.34 |
| | Llama-3.3-70B | 0.74 | 0.80 | 0.89 | 0.35 | 0.42 | 0.59 | 0.57 | 0.72 | 0.70 | 4.91 | 4.22 | 3.63 | 6.54 | 6.73 | 6.75 |
| | Qwen2.5-72B | 0.76 | 0.93 | 0.96 | 0.52 | 0.60 | 0.81 | 0.59 | 0.78 | 0.78 | 5.43 | 4.90 | 4.19 | 7.11 | 7.18 | 7.15 |
| | GPT-4o | 0.80 | 0.80 | 0.93 | 0.46 | 0.60 | 0.81 | 0.59 | 0.72 | 0.74 | 5.35 | 4.58 | 3.74 | 6.90 | 6.88 | 6.94 |
| | DeepSeek-V3.1 | 0.80 | 0.88 | 0.93 | 0.50 | 0.60 | 0.78 | 0.59 | 0.68 | 0.70 | 5.13 | 4.45 | 3.52 | 6.83 | 6.90 | 6.80 |
| | Gemini-2.5 | 0.93 | 0.93 | 0.93 | 0.59 | 0.62 | 0.70 | 0.67 | 0.72 | 0.74 | 5.35 | 4.58 | 3.48 | 6.96 | 6.67 | 6.70 |
| fisheries | Qwen2.5-32B | 0.87 | 0.86 | 0.89 | 0.54 | 0.58 | 0.48 | 0.74 | 0.77 | 0.66 | 5.43 | 4.91 | 4.16 | 6.77 | 6.93 | 6.64 |
| | Llama-3.3-70B | 0.88 | 0.89 | 0.84 | 0.54 | 0.62 | 0.61 | 0.72 | 0.74 | 0.68 | 5.45 | 4.92 | 4.34 | 7.01 | 7.13 | 6.83 |
| | Qwen2.5-72B | 0.93 | 0.89 | 0.86 | 0.80 | 0.58 | 0.48 | 0.84 | 0.74 | 0.68 | 6.45 | 4.74 | 4.27 | 7.54 | 6.99 | 6.71 |
| | GPT-4o | 0.94 | 0.97 | 0.95 | 0.72 | 0.74 | 0.77 | 0.81 | 0.83 | 0.77 | 6.28 | 5.55 | 4.98 | 7.33 | 7.52 | 7.35 |
| | DeepSeek-V3.1 | 0.96 | 0.92 | 0.93 | 0.78 | 0.74 | 0.75 | 0.77 | 0.83 | 0.77 | 6.20 | 5.52 | 4.93 | 7.30 | 7.45 | 6.94 |
| | Gemini-2.5 | 0.93 | 0.97 | 0.98 | 0.77 | 0.68 | 0.82 | 0.83 | 0.95 | 0.91 | 6.51 | 5.43 | 5.50 | 7.53 | 7.54 | 7.29 |
| transport & tourism | Qwen2.5-32B | 0.72 | 0.84 | 0.75 | 0.43 | 0.49 | 0.35 | 0.62 | 0.60 | 0.57 | 4.74 | 3.85 | 3.45 | 6.47 | 6.54 | 6.34 |
| | Llama-3.3-70B | 0.67 | 0.84 | 0.80 | 0.45 | 0.53 | 0.53 | 0.53 | 0.56 | 0.65 | 5.00 | 4.25 | 3.58 | 6.44 | 6.71 | 6.55 |
| | Qwen2.5-72B | 0.90 | 0.96 | 0.90 | 0.62 | 0.67 | 0.60 | 0.72 | 0.75 | 0.75 | 5.81 | 4.87 | 4.25 | 7.03 | 7.08 | 7.02 |
| | GPT-4o | 0.84 | 0.95 | 0.88 | 0.57 | 0.60 | 0.57 | 0.67 | 0.65 | 0.70 | 5.29 | 4.56 | 3.62 | 6.80 | 6.85 | 6.73 |
| | DeepSeek-V3.1 | 0.90 | 0.98 | 0.88 | 0.52 | 0.64 | 0.60 | 0.64 | 0.71 | 0.72 | 5.69 | 4.55 | 3.90 | 6.84 | 6.87 | 6.73 |
| | Gemini-2.5 | 0.98 | 0.96 | 0.88 | 0.62 | 0.67 | 0.53 | 0.76 | 0.78 | 0.70 | 5.71 | 4.35 | 3.80 | 6.70 | 6.67 | 6.63 |
| industry & energy | Qwen2.5-32B | 0.76 | 0.83 | 0.90 | 0.37 | 0.42 | 0.47 | 0.54 | 0.66 | 0.63 | 4.52 | 3.92 | 3.69 | 6.23 | 6.50 | 6.56 |
| | Llama-3.3-70B | 0.76 | 0.86 | 0.86 | 0.41 | 0.45 | 0.59 | 0.55 | 0.60 | 0.65 | 4.52 | 3.98 | 3.82 | 6.34 | 6.77 | 6.82 |
| | Qwen2.5-72B | 0.77 | 0.86 | 0.88 | 0.41 | 0.43 | 0.53 | 0.55 | 0.60 | 0.61 | 4.35 | 3.78 | 3.65 | 6.21 | 6.52 | 6.60 |
| | GPT-4o | 0.92 | 0.91 | 0.92 | 0.56 | 0.60 | 0.69 | 0.62 | 0.63 | 0.69 | 4.77 | 4.29 | 4.16 | 6.73 | 6.92 | 6.94 |
| | DeepSeek-V3.1 | 0.94 | 0.92 | 0.90 | 0.55 | 0.60 | 0.67 | 0.66 | 0.71 | 0.76 | 4.86 | 4.18 | 3.98 | 6.59 | 6.87 | 6.99 |
| | Gemini-2.5 | 0.59 | 0.83 | 0.61 | 0.52 | 0.66 | 0.71 | 0.75 | 0.58 | 0.69 | 4.32 | 3.71 | 4.12 | 6.15 | 6.84 | 6.79 |
| Average | Qwen2.5-32B | 0.78 | 0.84 | 0.86 | 0.42 | 0.49 | 0.44 | 0.61 | 0.68 | 0.64 | 4.87 | 4.26 | 3.76 | 6.49 | 6.63 | 6.49 |
| | Llama-3.3-70B | 0.77 | 0.85 | 0.84 | 0.44 | 0.51 | 0.58 | 0.60 | 0.65 | 0.67 | 4.97 | 4.36 | 3.87 | 6.59 | 6.85 | 6.75 |
| | Qwen2.5-72B | 0.84 | 0.91 | 0.89 | 0.59 | 0.56 | 0.58 | 0.68 | 0.71 | 0.69 | 5.50 | 4.52 | 4.06 | 6.95 | 6.91 | 6.83 |
| | GPT-4o | 0.89 | 0.92 | **0.92** | 0.59 | 0.64 | **0.71** | 0.68 | 0.71 | 0.72 | 5.43 | **4.77** | 4.18 | **6.95** | **7.07** | **7.00** |
| | DeepSeek-V3.1 | **0.91** | **0.93** | 0.91 | 0.60 | 0.65 | **0.69** | 0.67 | 0.74 | 0.74 | **5.49** | 4.71 | 4.14 | 6.89 | 7.04 | 6.88 |
| | Gemini-2.5 | 0.84 | 0.92 | 0.83 | **0.63** | **0.66** | 0.69 | **0.76** | **0.76** | **0.76** | 5.46 | 4.52 | **4.31** | 6.82 | 6.97 | 6.87 |

The results in Table 12 demonstrate the performance differences of various LLMs on the *PoliCon* benchmark's Industry theme, which includes four sub-themes: agriculture, fisheries, transport & tourism, and industry, research & energy. The results indicate that the pass rates for agriculture and fisheries topics are generally higher, possibly due to the relatively clear stance conflicts in these traditional industry topics, making it easier to reach compromises. In contrast, the performance on transport and tourism topics is slightly weaker (e.g., Qwen2.5-32B scores only 0.35 on the 2/3M task), suggesting that when it comes to cross-regional resource allocation, LLMs struggle to effectively safeguard the interests of the most vulnerable parties, directly related to the complexity of multiple stakeholders. Notably, topics like industry, research & energy, which involve technological transformation and policy coordination, score the lowest in Rawls tasks, reflecting the limitations of LLMs in handling issues at the intersection of technology and policy.

The results in Table 13 focus on budget-related topics in *PoliCon*, including development, regional development, budget, and budgetary control. They reveal distinct performance differences of LLMs on fiscal topics. Regional development topics demonstrate the highest consensus-building ability, likely due to their involvement with specific infrastructure projects, where benefit distribution schemes are easier to quantify and compromise on. In contrast, pure budget allocation topics (such as the budget fine-grained topic) show the weakest performance, reflecting the difficulty LLMs face in balancing multiple demands in abstract fiscal rule-making. Notably, budget control topics perform relatively well in the Util task under two-party settings. This suggests that LLMs are more effective at reaching technical consensus when the objective is framed in terms of measurable efficiency, rather than resolving deeper ideological disagreements over political principles or distributive justice.

Table 13: Performance of different LLMs on *PoliCon*'s Budget topic. The values in square brackets indicate the range of each metric, and all metrics follow the principle that higher values are better. The background color of the table cells deepens as the performance improves. The blue color scheme represents metrics in the 0-1 range, while the red color scheme represents metrics in the 0-9 range.

| Topic | Model | SM [0-1] ↑ | | | 2/3M [0-1] ↑ | | | VP [0-1] ↑ | | | Rawls [0-9] ↑ | | | Util [0-9] ↑ | | |
|---|---|---|---|---|---|---|---|---|---|---|---|---|---|---|---|---|
| | | 2 | 4 | 6 | 2 | 4 | 6 | 2 | 4 | 6 | 2 | 4 | 6 | 2 | 4 | 6 |
| development | Qwen2.5-32B | 0.75 | 0.81 | 0.85 | 0.38 | 0.48 | 0.45 | 0.62 | 0.52 | 0.65 | 4.16 | 3.23 | 2.55 | 6.20 | 6.24 | 6.42 |
| | Llama-3.3-70B | 0.78 | 0.84 | 0.80 | 0.44 | 0.48 | 0.55 | 0.62 | 0.42 | 0.60 | 3.97 | 2.90 | 2.00 | 6.19 | 6.31 | 6.53 |
| | Qwen2.5-72B | 0.78 | 0.90 | 0.85 | 0.56 | 0.68 | 0.65 | 0.72 | 0.55 | 0.70 | 4.91 | 3.35 | 2.65 | 6.84 | 6.65 | 6.79 |
| | GPT-4o | 0.78 | 0.87 | 0.85 | 0.56 | 0.58 | 0.65 | 0.62 | 0.48 | 0.70 | 4.75 | 3.29 | 2.40 | 6.53 | 6.51 | 6.75 |
| | DeepSeek-V3.1 | 0.84 | 0.84 | 0.95 | 0.56 | 0.61 | 0.65 | 0.66 | 0.52 | 0.70 | 4.44 | 3.29 | 2.85 | 6.58 | 6.57 | 6.68 |
| | Gemini-2.5 | 0.84 | 0.87 | 0.85 | 0.56 | 0.55 | 0.65 | 0.72 | 0.48 | 0.70 | 4.72 | 3.55 | 2.30 | 6.58 | 6.49 | 6.52 |
| regional development | Qwen2.5-32B | 0.85 | 0.93 | 0.89 | 0.48 | 0.52 | 0.57 | 0.60 | 0.62 | 0.71 | 4.90 | 3.64 | 3.17 | 6.68 | 6.53 | 6.75 |
| | Llama-3.3-70B | 0.79 | 0.93 | 0.89 | 0.56 | 0.60 | 0.63 | 0.58 | 0.64 | 0.66 | 4.92 | 3.64 | 2.97 | 6.76 | 6.75 | 6.91 |
| | Qwen2.5-72B | 0.90 | 0.98 | 0.94 | 0.69 | 0.79 | 0.74 | 0.67 | 0.69 | 0.77 | 5.65 | 4.21 | 3.63 | 7.14 | 7.08 | 7.16 |
| | GPT-4o | 0.90 | 0.95 | 0.91 | 0.67 | 0.76 | 0.63 | 0.69 | 0.71 | 0.86 | 5.27 | 4.07 | 3.63 | 7.01 | 6.86 | 7.02 |
| | DeepSeek-V3.1 | 0.94 | 0.95 | 1.00 | 0.69 | 0.62 | 0.66 | 0.75 | 0.69 | 0.83 | 5.65 | 4.26 | 3.66 | 7.11 | 6.86 | 7.05 |
| | Gemini-2.5 | 0.96 | 0.98 | 0.94 | 0.71 | 0.74 | 0.57 | 0.73 | 0.69 | 0.74 | 5.73 | 4.21 | 3.91 | 6.96 | 6.82 | 6.96 |
| budget | Qwen2.5-32B | 0.67 | 0.83 | 0.79 | 0.24 | 0.32 | 0.21 | 0.35 | 0.59 | 0.49 | 3.97 | 3.35 | 2.43 | 5.96 | 6.17 | 5.93 |
| | Llama-3.3-70B | 0.67 | 0.79 | 0.81 | 0.29 | 0.38 | 0.35 | 0.35 | 0.56 | 0.51 | 3.76 | 3.06 | 2.27 | 5.82 | 6.35 | 6.10 |
| | Qwen2.5-72B | 0.75 | 0.85 | 0.80 | 0.29 | 0.44 | 0.29 | 0.38 | 0.62 | 0.52 | 3.65 | 3.15 | 2.39 | 5.78 | 6.22 | 6.00 |
| | GPT-4o | 0.83 | 0.91 | 0.91 | 0.43 | 0.57 | 0.49 | 0.42 | 0.69 | 0.59 | 4.23 | 3.59 | 2.71 | 6.22 | 6.61 | 6.39 |
| | DeepSeek-V3.1 | 0.85 | 0.92 | 0.91 | 0.44 | 0.60 | 0.49 | 0.49 | 0.67 | 0.60 | 4.15 | 3.37 | 2.88 | 6.17 | 6.64 | 6.44 |
| | Gemini-2.5 | 0.95 | 0.90 | 0.71 | 0.45 | 0.56 | 0.40 | 0.58 | 0.71 | 0.61 | 4.47 | 3.41 | 2.61 | 6.22 | 6.45 | 6.28 |
| budgetary control | Qwen2.5-32B | 0.80 | 0.90 | 0.96 | 0.32 | 0.36 | 0.47 | 0.43 | 0.59 | 0.68 | 4.27 | 4.02 | 3.96 | 6.07 | 6.38 | 6.60 |
| | Llama-3.3-70B | 0.74 | 0.84 | 0.93 | 0.31 | 0.46 | 0.55 | 0.41 | 0.57 | 0.69 | 4.04 | 3.79 | 3.74 | 6.05 | 6.48 | 6.80 |
| | Qwen2.5-72B | 0.78 | 0.90 | 0.96 | 0.32 | 0.44 | 0.58 | 0.43 | 0.58 | 0.70 | 3.85 | 3.57 | 3.54 | 5.96 | 6.44 | 6.76 |
| | GPT-4o | 0.91 | 0.95 | 0.99 | 0.48 | 0.59 | 0.75 | 0.48 | 0.65 | 0.74 | 4.55 | 4.14 | 4.05 | 6.41 | 6.74 | 7.09 |
| | DeepSeek-V3.1 | 0.94 | 0.97 | 0.99 | 0.51 | 0.59 | 0.75 | 0.57 | 0.70 | 0.78 | 4.53 | 4.02 | 3.91 | 6.40 | 6.73 | 7.03 |
| | Gemini-2.5 | 0.95 | 0.97 | 0.98 | 0.57 | 0.61 | 0.70 | 0.65 | 0.73 | 0.76 | 4.86 | 4.36 | 4.15 | 6.43 | 6.67 | 6.96 |
| Average | Qwen2.5-32B | 0.78 | 0.88 | 0.93 | 0.32 | 0.37 | 0.44 | 0.43 | 0.59 | 0.65 | 4.25 | 3.83 | 3.67 | 6.10 | 6.34 | 6.52 |
| | Llama-3.3-70B | 0.73 | 0.84 | 0.91 | 0.33 | 0.45 | 0.53 | 0.42 | 0.57 | 0.66 | 4.04 | 3.60 | 3.44 | 6.06 | 6.46 | 6.71 |
| | Qwen2.5-72B | 0.78 | 0.89 | 0.93 | 0.35 | 0.47 | 0.55 | 0.45 | 0.59 | 0.68 | 3.99 | 3.52 | 3.37 | 6.05 | 6.45 | 6.69 |
| | GPT-4o | 0.89 | 0.94 | 0.97 | 0.48 | **0.60** | **0.71** | 0.49 | 0.65 | 0.73 | 4.54 | 3.99 | 3.79 | 6.42 | **6.71** | **6.99** |
| | DeepSeek-V3.1 | 0.92 | **0.96** | **0.98** | 0.51 | **0.60** | **0.71** | 0.57 | 0.69 | **0.76** | 4.53 | 3.87 | 3.73 | 6.41 | **6.71** | 6.94 |
| | Gemini-2.5 | **0.94** | 0.95 | 0.94 | **0.55** | 0.60 | 0.65 | **0.65** | **0.71** | 0.74 | **4.84** | **4.13** | **3.87** | **6.43** | 6.63 | 6.86 |

Table 14: Performance of different LLMs on *PoliCon*'s Security topic. The values in square brackets indicate the range of each metric, and all metrics follow the principle that higher values are better. The background color of the table cells deepens as the performance improves. The blue color scheme represents metrics in the 0-1 range, while the red color scheme represents metrics in the 0-9 range.

| Model | Topic | SM [0-1] ↑ | | | 2/3M [0-1] ↑ | | | VP [0-1] ↑ | | | Rawls [0-9] ↑ | | | Util [0-9] ↑ | | |
|---|---|---|---|---|---|---|---|---|---|---|---|---|---|---|---|---|
| | | 2 | 4 | 6 | 2 | 4 | 6 | 2 | 4 | 6 | 2 | 4 | 6 | 2 | 4 | 6 |
| public health | Qwen2.5-32B | 0.79 | 0.83 | 0.89 | 0.38 | 0.48 | 0.51 | 0.53 | 0.59 | 0.71 | 4.08 | 3.63 | 3.38 | 6.16 | 6.59 | 6.70 |
| | Llama-3.3-70B | 0.81 | 0.87 | 0.90 | 0.43 | 0.54 | 0.55 | 0.52 | 0.64 | 0.73 | 4.24 | 3.72 | 3.49 | 6.34 | 6.75 | 6.86 |
| | Qwen2.5-72B | 0.77 | 0.83 | 0.91 | 0.35 | 0.54 | 0.55 | 0.54 | 0.66 | 0.71 | 4.07 | 3.73 | 3.45 | 6.12 | 6.62 | 6.69 |
| | GPT-4o | 0.88 | 0.91 | 0.96 | 0.56 | 0.66 | 0.64 | 0.59 | 0.70 | 0.77 | 4.82 | 4.14 | 3.58 | 6.65 | 6.97 | 7.03 |
| | DeepSeek-V3.1 | 0.89 | 0.91 | 0.94 | 0.56 | 0.67 | 0.67 | 0.65 | 0.67 | 0.77 | 4.76 | 4.17 | 3.76 | 6.61 | 6.98 | 7.04 |
| | Gemini-2.5 | 0.94 | 0.93 | 0.94 | 0.62 | 0.67 | 0.60 | 0.65 | 0.71 | 0.76 | 5.01 | 4.21 | 3.38 | 6.53 | 6.83 | 6.96 |
| foreign & security | Qwen2.5-32B | 0.66 | 0.68 | 0.79 | 0.31 | 0.35 | 0.32 | 0.38 | 0.46 | 0.53 | 3.54 | 3.02 | 2.49 | 5.50 | 5.89 | 6.09 |
| | Llama-3.3-70B | 0.63 | 0.66 | 0.76 | 0.34 | 0.38 | 0.32 | 0.38 | 0.48 | 0.51 | 3.47 | 2.93 | 2.26 | 5.61 | 5.96 | 6.26 |
| | Qwen2.5-72B | 0.66 | 0.69 | 0.79 | 0.32 | 0.39 | 0.33 | 0.39 | 0.48 | 0.52 | 3.53 | 2.86 | 2.42 | 5.52 | 5.93 | 6.11 |
| | GPT-4o | 0.75 | 0.76 | 0.84 | 0.45 | 0.49 | 0.51 | 0.48 | 0.54 | 0.60 | 4.09 | 3.32 | 2.67 | 5.93 | 6.23 | 6.46 |
| | DeepSeek-V3.1 | 0.78 | 0.78 | 0.86 | 0.46 | 0.47 | 0.53 | 0.47 | 0.53 | 0.61 | 4.06 | 3.35 | 2.78 | 5.90 | 6.19 | 6.41 |
| | Gemini-2.5 | 0.80 | 0.80 | 0.84 | 0.40 | 0.47 | 0.46 | 0.41 | 0.57 | 0.62 | 4.23 | 3.43 | 2.87 | 5.95 | 6.18 | 6.33 |
| Average | Qwen2.5-32B | 0.70 | 0.72 | 0.83 | 0.33 | 0.39 | 0.39 | 0.43 | 0.50 | 0.60 | 3.70 | 3.21 | 2.83 | 5.70 | 6.10 | 6.32 |
| | Llama-3.3-70B | 0.69 | 0.73 | 0.81 | 0.37 | 0.43 | 0.41 | 0.42 | 0.53 | 0.59 | 3.70 | 3.18 | 2.73 | 5.83 | 6.21 | 6.49 |
| | Qwen2.5-72B | 0.70 | 0.74 | 0.84 | 0.33 | 0.44 | 0.41 | 0.44 | 0.54 | 0.59 | 3.69 | 3.14 | 2.81 | 5.70 | 6.15 | 6.33 |
| | GPT-4o | 0.79 | 0.80 | **0.89** | 0.48 | **0.54** | 0.56 | 0.51 | **0.59** | 0.67 | 4.31 | 3.57 | 3.01 | **6.15** | **6.46** | **6.68** |
| | DeepSeek-V3.1 | 0.81 | 0.82 | **0.89** | **0.49** | **0.54** | **0.58** | **0.52** | 0.58 | 0.67 | 4.27 | 3.61 | **3.15** | 6.12 | 6.44 | 6.65 |
| | Gemini-2.5 | **0.84** | **0.84** | 0.88 | 0.47 | 0.53 | 0.52 | 0.49 | **0.61** | **0.67** | **4.46** | **3.67** | 3.07 | 6.13 | 6.38 | 6.57 |

Table 14 illustrates the significant differences among various LLMs on two fine-grained topics under the Security theme: environment & public health and foreign & security policy. The environment & public health topic demonstrates a higher consensus-building ability than foreign & security policy across all five task settings, likely due to its technical and non-political nature, which allows models to reconcile different positions more easily. In contrast, the foreign & security policy topic performs bad across all task settings, highlighting the limitations of LLMs when handling highly sensitive issues like national sovereignty and geopolitics. Notably, in the Rawls task, the environment & public health topic scores significantly higher than foreign & security policy, indicating that LLMs achieve better consensus in healthcare fields, while struggling to overcome established power structures in complex political issues related to national security. This disparity supports the conclusion throughout the text regarding how topic complexity affects model performance, especially with the value conflicts and zero-sum nature unique to security topics.

Table 15: Performance of different LLMs on *PoliCon*'s Civil Rights topic. The values in square brackets indicate the range of each metric, and all metrics follow the principle that higher values are better. The background color of the table cells deepens as the performance improves. The blue color scheme represents metrics in the 0-1 range, while the red color scheme represents metrics in the 0-9 range.

| Topic | Model | SM [0-1] ↑ | | | 2/3M [0-1] ↑ | | | VP [0-1] ↑ | | | Rawls [0-9] ↑ | | | Util [0-9] ↑ | | |
|---|---|---|---|---|---|---|---|---|---|---|---|---|---|---|---|---|
| | | 2 | 4 | 6 | 2 | 4 | 6 | 2 | 4 | 6 | 2 | 4 | 6 | 2 | 4 | 6 |
| culture & education | Qwen2.5-32B | 0.88 | 0.86 | 0.92 | 0.49 | 0.50 | 0.69 | 0.46 | 0.47 | 0.73 | 3.71 | 2.81 | 2.69 | 6.04 | 6.28 | 6.67 |
| | Llama-3.3-70B | 0.83 | 0.86 | 0.96 | 0.51 | 0.56 | 0.73 | 0.51 | 0.47 | 0.69 | 3.56 | 2.78 | 2.65 | 6.11 | 6.44 | 6.79 |
| | Qwen2.5-72B | 0.90 | 0.89 | 0.96 | 0.63 | 0.58 | 0.73 | 0.59 | 0.53 | 0.77 | 4.49 | 3.11 | 3.19 | 6.49 | 6.70 | 7.12 |
| | GPT-4o | 0.90 | 0.89 | 0.92 | 0.56 | 0.64 | 0.81 | 0.59 | 0.50 | 0.69 | 4.34 | 3.00 | 3.19 | 6.51 | 6.53 | 6.90 |
| | DeepSeek-V3.1 | 0.95 | 0.97 | 0.96 | 0.61 | 0.64 | 0.81 | 0.61 | 0.58 | 0.77 | 4.32 | 3.36 | 3.27 | 6.50 | 6.62 | 6.93 |
| | Gemini-2.5 | 0.88 | 0.89 | 0.88 | 0.63 | 0.56 | 0.69 | 0.59 | 0.61 | 0.73 | 4.78 | 3.36 | 3.19 | 6.45 | 6.53 | 6.74 |
| gender equality | Qwen2.5-32B | 0.51 | 0.73 | 0.93 | 0.32 | 0.45 | 0.56 | 0.36 | 0.41 | 0.59 | 2.85 | 2.48 | 2.11 | 5.85 | 6.29 | 6.49 |
| | Llama-3.3-70B | 0.49 | 0.75 | 0.89 | 0.38 | 0.45 | 0.59 | 0.28 | 0.45 | 0.63 | 2.83 | 2.34 | 2.26 | 5.87 | 6.24 | 6.46 |
| | Qwen2.5-72B | 0.60 | 0.75 | 0.93 | 0.36 | 0.48 | 0.56 | 0.36 | 0.48 | 0.63 | 3.26 | 2.82 | 2.33 | 5.98 | 6.38 | 6.69 |
| | GPT-4o | 0.57 | 0.77 | 0.93 | 0.34 | 0.48 | 0.59 | 0.30 | 0.43 | 0.63 | 3.11 | 2.70 | 2.48 | 5.93 | 6.28 | 6.54 |
| | DeepSeek-V3.1 | 0.60 | 0.80 | 0.89 | 0.34 | 0.48 | 0.52 | 0.34 | 0.43 | 0.63 | 3.43 | 3.02 | 2.70 | 5.90 | 6.37 | 6.61 |
| | Gemini-2.5 | 0.74 | 0.80 | 0.93 | 0.38 | 0.43 | 0.56 | 0.43 | 0.45 | 0.63 | 3.32 | 2.86 | 2.56 | 5.91 | 6.26 | 6.52 |
| civil liberties | Qwen2.5-32B | 0.64 | 0.72 | 0.78 | 0.24 | 0.31 | 0.27 | 0.43 | 0.47 | 0.58 | 3.18 | 2.80 | 2.60 | 5.64 | 5.89 | 6.05 |
| | Llama-3.3-70B | 0.66 | 0.66 | 0.81 | 0.27 | 0.39 | 0.39 | 0.44 | 0.45 | 0.59 | 3.20 | 2.70 | 2.55 | 5.73 | 5.99 | 6.18 |
| | Qwen2.5-72B | 0.66 | 0.68 | 0.81 | 0.29 | 0.38 | 0.34 | 0.42 | 0.46 | 0.59 | 3.27 | 2.70 | 2.39 | 5.65 | 5.88 | 6.05 |
| | GPT-4o | 0.76 | 0.77 | 0.88 | 0.43 | 0.46 | 0.56 | 0.53 | 0.58 | 0.66 | 3.86 | 3.10 | 2.94 | 6.00 | 6.22 | 6.52 |
| | DeepSeek-V3.1 | 0.85 | 0.84 | 0.89 | 0.44 | 0.48 | 0.57 | 0.56 | 0.57 | 0.65 | 3.92 | 3.21 | 2.94 | 6.05 | 6.22 | 6.55 |
| | Gemini-2.5 | 0.71 | 0.86 | 0.89 | 0.32 | 0.44 | 0.50 | 0.53 | 0.59 | 0.67 | 1.99 | 3.48 | 3.07 | 6.12 | 6.26 | 6.46 |
| constitutional affairs | Qwen2.5-32B | 0.61 | 0.69 | 0.68 | 0.21 | 0.20 | 0.23 | 0.42 | 0.54 | 0.59 | 2.98 | 2.46 | 2.18 | 5.32 | 5.82 | 5.85 |
| | Llama-3.3-70B | 0.67 | 0.69 | 0.68 | 0.35 | 0.30 | 0.27 | 0.44 | 0.48 | 0.59 | 3.07 | 2.46 | 2.14 | 5.53 | 5.89 | 6.02 |
| | Qwen2.5-72B | 0.82 | 0.83 | 0.84 | 0.53 | 0.50 | 0.41 | 0.60 | 0.63 | 0.70 | 3.51 | 2.83 | 2.57 | 6.04 | 6.25 | 6.39 |
| | GPT-4o | 0.81 | 0.89 | 0.86 | 0.44 | 0.44 | 0.43 | 0.51 | 0.61 | 0.68 | 3.16 | 2.70 | 2.50 | 5.75 | 6.13 | 6.23 |
| | DeepSeek-V3.1 | 0.79 | 0.83 | 0.86 | 0.42 | 0.46 | 0.45 | 0.58 | 0.63 | 0.61 | 3.23 | 2.78 | 2.23 | 5.81 | 6.17 | 6.25 |
| | Gemini-2.5 | 0.89 | 0.87 | 0.89 | 0.42 | 0.50 | 0.41 | 0.61 | 0.67 | 0.61 | 3.47 | 2.83 | 2.43 | 5.90 | 6.19 | 6.09 |
| legal affairs | Qwen2.5-32B | 0.75 | 0.84 | 0.92 | 0.39 | 0.51 | 0.46 | 0.51 | 0.63 | 0.70 | 4.20 | 3.69 | 3.35 | 6.19 | 6.54 | 6.64 |
| | Llama-3.3-70B | 0.76 | 0.78 | 0.89 | 0.37 | 0.53 | 0.43 | 0.49 | 0.59 | 0.70 | 4.42 | 3.92 | 3.32 | 6.35 | 6.65 | 6.75 |
| | Qwen2.5-72B | 0.86 | 0.84 | 0.95 | 0.53 | 0.69 | 0.70 | 0.63 | 0.67 | 0.76 | 5.25 | 4.65 | 3.86 | 6.86 | 7.07 | 7.24 |
| | GPT-4o | 0.83 | 0.88 | 0.95 | 0.56 | 0.63 | 0.59 | 0.59 | 0.69 | 0.73 | 5.10 | 4.39 | 3.76 | 6.74 | 6.90 | 7.07 |
| | DeepSeek-V3.1 | 0.86 | 0.92 | 0.95 | 0.53 | 0.69 | 0.62 | 0.63 | 0.71 | 0.78 | 5.14 | 4.29 | 3.89 | 6.75 | 6.92 | 7.00 |
| | Gemini-2.5 | 0.93 | 0.80 | 0.95 | 0.64 | 0.49 | 0.57 | 0.68 | 0.76 | 0.73 | 5.61 | 4.59 | 4.03 | 6.69 | 6.84 | 6.84 |
| Average | Qwen2.5-32B | 0.66 | 0.75 | 0.81 | 0.30 | 0.36 | 0.37 | 0.44 | 0.50 | 0.62 | 3.33 | 2.84 | 2.59 | 5.75 | 6.07 | 6.22 |
| | Llama-3.3-70B | 0.67 | 0.71 | 0.82 | 0.34 | 0.42 | 0.44 | 0.44 | 0.48 | 0.62 | 3.37 | 2.80 | 2.57 | 5.86 | 6.16 | 6.34 |
| | Qwen2.5-72B | 0.74 | 0.76 | 0.86 | 0.41 | 0.48 | 0.47 | 0.49 | 0.53 | 0.66 | 3.76 | 3.07 | 2.73 | 6.04 | 6.28 | 6.48 |
| | GPT-4o | 0.77 | 0.82 | 0.90 | 0.46 | 0.51 | 0.57 | 0.51 | 0.57 | 0.67 | 3.91 | 3.16 | 2.96 | 6.13 | 6.35 | 6.59 |
| | DeepSeek-V3.1 | 0.82 | 0.86 | 0.90 | 0.46 | 0.53 | 0.58 | 0.55 | 0.58 | 0.67 | 3.99 | 3.29 | 2.96 | 6.16 | 6.38 | 6.61 |
| | Gemini-2.5 | 0.80 | 0.84 | 0.90 | 0.43 | 0.47 | 0.52 | 0.56 | 0.61 | 0.67 | 3.30 | 3.44 | 3.06 | 6.19 | 6.36 | 6.49 |

Table 15 focuses on these five subtopics: culture & education, gender equality, civil liberties, justice & home affairs, constitutional & inter-institutional affairs, and legal affairs. These fine-grained topics reveal significant differences in how LLMs handle various Civil Rights issues. The topic of legal affairs shows the strongest consensus-building ability, possibly because it is grounded in established legal frameworks, precedents, and procedural norms that provide clear reference points for reasoning. In contrast, the topic of constitutional affairs performs the weakest, reflecting the difficulty LLMs face

in overcoming opposing stances when fundamental constitutional principles are involved. Notably, the civil liberties, justice & home affairs topic exhibits the most fluctuation in scores on the 2/3M task (Qwen2.5-32B scores only 0.24 while DeepSeek-V3.1 reaching 0.57), indicating that this issue is the most sensitive to the models' value orientations.

## G    Discussion and Limitations

As discussed in section 5, although *PoliCon* has demonstrated an excellent ability to evaluate LLMs in achieving different objectives of political consensus, we acknowledge that it still faces some limitations. In this section, we will discuss these in more detail.

Firstly, we introduced LLMs into the data cleaning process. Although our implementation was designed to minimize bias as much as possible, some residual influence may still remain. Secondly, AI-generated content that carries risks or offensive language toward certain groups may have a potentially negative impact on the question of using AI to advance political objectives. Additionally, there is a risk of data leakage in our dataset. However, not only have we mitigated this effect by setting task configurations different from the real world, but our experiments also show that current state-of-the-art LLMs are not very effective at handling tasks that involve finding political consensus across different tasks. This suggests that the impact of data leakage might not be significant.

In the future, since generating task scenarios incurs no cost, we can customize a large number of test scenarios flexibly and diversely according to specific needs. This can further enable our work to be applied to broader research settings, such as Pareto improvements and multi-objective optimization research, as well as research on different deliberation algorithms, and our evaluation framework can even retain its algorithm-agnostic feature, which can also be considered in future work.

## H    Ethical Statement and Disclaimer

In this section, we will discuss the copyright issues of the data sources in this paper, the potential social risks, and the statement regarding the proper use of the data in *PoliCon*.

### H.1    Copyright of Data Sources

The data in this paper is sourced and organized from the official website of the European Parliament[29], HowTheyVote[30], and the VoteWatch Europe dataset (HIX et al., 2022). Both the official website of the European Parliament and HowTheyVote allow the use of their data as long as the source is cited, while the VoteWatch Europe dataset follows the CC 4.0 license.

### H.2    Potential Societal Impact and Statement on the Use of *PoliCon*

*PoliCon*, as an AI project with the potential to influence social governance processes, carries certain social risks. For instance, it might generate biased or offensive statements towards specific groups when producing consensus decisions. Additionally, the use of AI systems in social governance processes could have both short-term and long-term impacts. Short-term effects might include generating persuasive rhetoric or exploiting cognitive biases of government officials, such as the anchoring effect, thereby reinforcing legislators' existing biases. It could also lead to legislators becoming overly reliant on automated tools, neglecting more comprehensive research, consultation, and deliberation. In the long term, it might amplify social issues, lock in certain values and knowledge, or lead to unpredictable risks and adverse outcomes. Before applying it to real-world governance processes, it is crucial to extensively consider its potential social risks.

The data in *PoliCon* has undergone processing using LLMs, including filtering, summarizing, and translating, as well as expanded settings for specific tasks, such as adjusting the distribution of seats among different parties and adding additional voting rules. During the LLM data processing, although the content is directly related to the original text, inherent biases and harmful statements may still be introduced from the LLMs. Additionally, we do not rule out the possibility of omissions during data

---

[29]https://www.europarl.europa.eu
[30]https://howtheyvote.eu

collection. These factors mean that our benchmark does not necessarily have a direct correlation with real-world European Parliament decisions and cannot be used to represent or predict any political outcomes or statements of the European Parliament.

It is worth noting that *PoliCon* should be only used for scientific research and academic purposes. If any third party uses *PoliCon* to make inappropriate statements, actions, or harmful legal suggestions regarding political, ethical, or other issues, this paper is not responsible for such actions. Additionally, since the data sources of *PoliCon* are real parliamentary data, they may contain politically sensitive statements from certain countries and regions, which do not represent any political views of the authors of this article.

## I   THE USAGE OF LARGE LANGUAGE MODELS

This paper employed LLMs for grammar checking and stylistic refinement during the writing process. In the data processing stage, LLMs were used to refine and filter the raw data. We further adopted an LLM-as-a-judge approach to build a sub-module of our evaluation framework, which simulates the real voting outcomes of each political party. Finally, since this work is positioned as an LLM benchmark study, LLMs themselves serve as the objects of investigation.

