# OpenReview forum: "PoliCon: Evaluating LLMs on Achieving Diverse Political Consensus Objectives"
_ICLR.cc/2026/Conference — ICLR 2026 Poster_

### Official Review · Reviewer_u21o · 2025-10-25

**Soundness:** 2
**Presentation:** 2
**Contribution:** 3
**Rating:** 6
**Confidence:** 4

**Summary:**

The authors propose a benchmark to evaluate whether LLMs can draft consensus resolutions under realistic political-committee constraints. The dataset is built from European Parliament records (2009–2022) and, per the paper, comprises 2,225 cleaned issues, each with (issue, topic, debates, resolution, votes). Each evaluation scenario specifies: (1) a topic (5 coarse / 19 fine categories), (2) a political goal (passing a resolution with different voting rules; Rawlsianism; Utilitarianism), (3) participating parties, and (4) a power structure (randomized seat shares and, optionally, a veto party).
Six LLMs are benchmarked. The benchmarked models often succeed in simple-majority settings but struggle with two-thirds, veto, security topics, and Rawls. They also exhibit partisan biases aligned with real EP voting patterns. The authors claim the judge correlates with “ground-truth” party votes with Pearson 0.83, and >72% of simulations fall within 2 SD of ground truth.

**Strengths:**

1. Evaluating LLMs on stakeholder consensus formation (with explicit power structures and collective-choice objectives) is a significant step beyond standard dialogue or persuasion tasks
2. Tying tasks to EP issues, debates, and resolutions is realistic (and much better than synthetic setups)
3. The topic taxonomy is broad and policy-relevant
4. Detailed scraping pipelines, prompt templates, and task descriptions are provided (high evaluation transparency)
5. Multiple models, tasks, and topic analyses expose nuanced failure modes (e.g., difficulty with Security and Rawls and specific biases)

**Weaknesses:**

1. LLM-as-judge with same-family system under test (GPT-4o judge vs GPT-4o candidate) risks systematic biases. A human (expert) verification, cross-judge, or calibration would be helpful.
2. The core of the aggregation, i.e., u_i = JUDGE(· | background, s_i, resolution) remains undefined (clearly under-specified). How exactly is alignment and feasibility integrated into this score?
3. It is not clear how the SD in Fig. 4 is exactly calculated. Can you please provide the exact error formula you are using?
4. Obviously, some scenarios have been omitted from the analyses. Can you please state how exactly you proceeded in the selection phase? In addition, you state that there were 2,225 "high-quality", i.e., complete records, but in Table 6 (last row), you report more than 2,700 total valid records. Can you please clarify?
5. Your vetoing rule seems to be contradictory. Does rejection by the vetoing party happen if support is under 60% or at least 60%? Please clarify.
6. DeepSeek-R1 is used to summarize, clean, and to extract party stances. How do you avoid errors or potential hallucinations? Human-based quality assessment, preferably with inter-annotator checks would be helpful.
7. It is not clear how the number of seats per party affects the veto robustness. For each issue, you could sample 20 to 50 random seat distributions (and veto assignments where applicable) and report the mean with CI per model and task.
8. It is difficult to put the LLM performance into relation to a baseline because there is no simple (i.e., interpretable) baseline in the paper. Would it be possible to provide such a baseline? For example, if a proposal fails to meet the threshold (e.g., simple majority, 2/3), iteratively sacrifice the interests of the party that contributes least to passing (i.e., with smallest w_i). This would be a greedy strategy to pass the threshold.

**Questions:**

See comments above.

---

> ### Author Response · Authors · 2025-11-20
>
> Thank you, reviewer u21o, for your constructive feedback. Here are our responses, organized to address each of your concerns one by one, specifically:
>
> > **Q1. Human verification is needed for the LLM-as-a-Judge-based evaluator.**
>
> **A1**: Thank you for your question. We have incorporated a detailed human study to address your concern. Because our task involves both AI and political science, we recruit 20 human annotators. Half of them have AI-related backgrounds, and the other half are law students, allowing the team to cover a broader range of relevant domain knowledge.
>
> We randomly sample 100 consensus resolutions generated by LLMs for evaluation. To ensure annotation robustness, each data item is assigned to three different annotators, yielding a total of 300 annotation results. Consistent with the evaluator's scoring method, we use the expected values of consistency and feasibility as the final evaluation scores. We present the consistency results of our evaluator compared with the ground truth and human evaluations in the following table:
>
> **Consistency of our evaluator with ground truth and human evaluations.**
> |                | Ground truth | Human eval. |
> |:----------------:|:--------------:|:-------------:|
> | Mean error     | 1.36         | 1.61        |
> | σ              | 1.90         | 1.92        |
> | Within ±σ      | 72%          | 72%         |
>
> We report the mean error, the standard deviation $\sigma$, and the proportion of samples whose errors fall within $\pm \sigma$. It can be seen that the mean error between our evaluator and human annotations is only 1.61, and more than 72% of the data errors fall within $\pm 1.92$, which demonstrates that our evaluator is sufficient to generalize in evaluating LLM-generated consensus resolutions. Another interesting phenomenon is that the consistency between our evaluator and human annotators is highly aligned with the consistency between our evaluator and the real-world ground-truth voting results, indicating that our consistency-computation method based on real voting data is also effective.
>
> This part has been detailed in lines 371-416 of Section 4.1 and the newly added Appendix E.2 in our revised manuscript.
>
> > **Q2. How are alignment and feasibility actually integrated into the scoring for each stakeholder?**
>
> **A2**: Thank you for your question. Regarding the calculation method, we use two separate prompts to have the judge model score alignment and feasibility, respectively. The specific prompts are detailed in Appendix C.3. After obtaining these two scores, we take their average as the final score for $u_i$. We have added a description of this process in line 1627 of Appendix C.3 in our revised manuscript.

---

> ### Author Response · Authors · 2025-11-20
>
> > **Q3. Could you provide the specific error calculation formula that you used in Figure 4?**
>
> **A3**: Thank you for your question. We primarily use gaussian_kde from scipy.stats to calculate the errors. Specifically, the complete calculation process can be divided into the following three steps: (1) compute the errors; (2) compute the standard deviation of the errors; (3) use Gaussian kernel density estimation (Gaussian KDE) to obtain the probability density function and plot it. Next, I will provide a detailed introduction to the calculation formulas and procedures for each step:
>
> **1. Definition and Calculation of Error**: We define the error as the difference between the score predicted by our judge model and the real-world ground truth. Formally, let there be a total of $n$ observed samples, and denote the set of sample indices as $\mathcal{I} = {1,2,\dots,n}$. For each $i \in \mathcal{I}$, let $p_i \in [0,9] \cap \mathbb{N}$ be the predicted score and $g_i \in [0,9] \cap \mathbb{N}$ be the corresponding ground-truth score, where $\mathbb{N}$ denotes the set of natural numbers. Then, the prediction error for the $i$-th sample is defined as: $e_i = p_i - g_i$.
>
> **2. Standard deviation of the error**: We first calculate the mean of the errors $\bar e = \frac{1}{n} \sum_{i=1}^n e_i$, and then compute its standard deviation $\sigma = \sqrt{\frac{1}{n} \sum_{i=1}^n (e_i - \bar e)^2}$.
>
> **3. Obtain the probability density function using Gaussian KDE and plot it**: Gaussian Kernel Density Estimation is a method that treats each sample point as the center of a Gaussian distribution and obtains a mixture of Gaussian distributions to estimate the probability density function of a random variable. It is usually defined by the following formula:
> $\hat f_h(x) = \frac{1}{n h \sqrt{2\pi}} \sum_{i=1}^n \exp \left( - \frac{1}{2} \left( \frac{x - e_i}{h} \right)^2 \right)$,
>
> where $h$ is the smoothing parameter. In this work, we implement this function using gaussian_kde from scipy.stats, and the selection of $h$ is automatically determined by the Scott method [1] in gaussian_kde.
>
> Finally, we sample $x \in [-9, 9]$ with an interval of 0.001 to compute the values, thereby obtaining the error probability density curve in Figure 4. We have provided a detailed description of this procedure in line 362 of the revised manuscript and in the newly added Appendix E.1.

---

> ### Author Response · Authors · 2025-11-20
>
> > **Q4. What were your specific steps during the filtering phase? Additionally, you mentioned that there were 2,225 complete records, but Table 6 shows that the total number of valid records exceeds 2,700. Could you please explain this?**
>
> **A4**: Your observation is very careful, and we sincerely thank you for pointing this out, which is extremely helpful for improving our work! Upon inspection, we found that we only introduced the filtering process for the raw text in Appendix A.5, but in Appendix A.6, we missed the introduction of the two filtering steps applied after the data cleaning process through DeepSeek-R1. To explain more clearly, let me first introduce our complete data filtering process: (1) Remove duplicate records in the raw data records, (2) Remove records with missing debate texts. After the data cleaning process, we perform: (3) Remove records with missing resolution or stance texts, and (4) Categorize the final records by topic and filter out topics with too few records. We have already described the first two steps in Appendix A.5. Next, let me provide a detailed explanation of the third and fourth steps. To better demonstrate that these steps were indeed missed in our appendix, we will include the core code used to process these two steps for your reference.
>
> For the third step, our operation is first to simply remove records where the resolution is an empty string. The core code is as follows:
> ```python
> topic_data_nums = {}
> topic_datas = {}
> for data in datas:
>     if data["resolution"] != "" and len(data["debate"]) > 0  and len(data["debate"]["views"]) > 0:
>         topic_data_nums[data['topic']] += 1
>         topic_datas[data['topic']].append(data)
> ```
> The situation is slightly more complex for missing stances, because sometimes the stances string is not empty but is None or contains fewer than five words. Therefore, we use the following rule-based method to filter the original data with missing stances. If all stances of a record are filtered in the end, the record itself is filtered. The core code is as follows:
> ```python
> # For removing the invalid stances
> for topic_name in topic_data_nums.keys():
> 	for data in topic_datas[topic_name]:
> 	    tmp_stances_datas = []
> 	    for stance in data['stances']:
> 	        if 'None' not in stance['stance'][:10] and stance['stance'] != '' and len(stance['stance']) > 50:
> 	            tmp_stances_datas.append(stance)
> 	    data['stances'] = tmp_stances_datas
>
> # For removing the issue with no stances left
> for topic_name in topic_data_nums.keys():
> 	tmp_topic_data = []
> 	for data in topic_datas[topic_name]:
> 		party_num = len(data['stances'])
> 		if party_num != 0:
> 			tmp_topic_data.append(data)
> 		else:
> 			topic_data_nums[topic_name] -= 1
> 	topic_datas[topic_name] = tmp_topic_data
> ```
> In this process, removing empty resolutions filtered out 99 records, and removing empty stances filtered out 375 records. Therefore, a total of 99 + 375 = 474 records were filtered out in this step.
>
> For the fourth step, since some topics have very few records, for example, the topic "juridical affairs" contains only 13 records, treating it as an independent task would result in insufficient data. Therefore, we removed topics with 25 records or fewer. The core code used for this is as follows:
> ```python
> for topic_name in topic_data_nums.keys():
>     if topic_data_nums[topic_name] > 25:
>         save_json(topic_datas[topic_name], f"{topic_name}.json")
> ```
> In this step, a total of two topics were removed: "juridical affairs" (only 13 records) and "petitions" (only 23 records), filtering a total of 36 records.
>
> Therefore, in the last two steps, we filtered a total of 474 + 36 = 510 records. After these two steps, the total number of records decreased from the original 2,735 to the final 2,225. This part has been supplemented in our revised manuscript of Appendix A.6.
>
> > **Q5. Your rule on the task of veto power seems contradictory. Does the vetoing party reject a proposal when the support rate is below 60%, or when the support rate reaches or exceeds 60%?**
>
> **A5**: Thank you for your question. In lines 301–305, we define the task setting for the veto power, which requires that the consensus proposal can be passed by a simple majority only if the in-favor rate of the vetoing party is not less than 60%. That is, in this setting, $v = 1$ only if $u \geq 5$ and $u_k \geq 6$, where $u_k$ is the voting score of the vetoing party. However, we realize that our wording may have caused this misunderstanding, so in the revised version, we have modified line 303 to remove any ambiguity in interpretation.

---

> ### Author Response · Authors · 2025-11-20
>
> > **Q6. Is it possible to incorporate human quality evaluation in the data cleaning process of DeepSeek-R1?**
>
> **A6**: Thank you for your question. We have conducted a human study to assess the impact of LLMs on the quality of data cleaning. We recruit 10 human annotators and design an intuitive and easy-to-use interface for the experiment: the interface displays the original and cleaned resolutions and stances, and annotators are asked to evaluate the quality of the LLM-based data cleaning for each sample using a scalar score from 0 to 5. We randomly sample 100 instances from the dataset and assign 10 instances to each annotator as their labeling task. The results show that human annotators rate more than 95.8\% of the cleaned resolution samples and more than 75\% of the cleaned stances samples at 3 points or above. This indicates that DeepSeek-R1 is highly effective in improving data-cleaning quality and achieves an acceptable level of subjective quality.
>
> The human study has been detailed in our revised manuscript in lines 1785–1794 of Appendix D, as well as in the newly added Appendix E.2.
>
> > **Q7. It is currently unclear how the number of seats held by each party affects the robustness of veto power. For each issue, you can randomly draw 20 to 50 samples of seat distributions and report the mean and confidence interval for each model and task.**
>
> **A7**: Thank you for your question. To examine how the number of seats held by each party affects the robustness of the task of veto power, we design two experiments. The first tests whether an increasing sample number in each party's seats strengthens or weakens the robustness of veto power; the second tests how strongly different party seat allocations influence the robustness of the final outcome.
>
> **Experiment 1**: Since we randomly draw one seat allocation for each data point, computing each model's overall expected outcome over a large set of data with varying seat distributions is clearly a robust way to test the model's veto power. Therefore, if in an experiment where we sample multiple different numbers of seats for each issue and compute the corresponding expected outcomes, the resulting experimental findings are consistent with the performance of the models presented in our Table 1, this would indicate that increasing the number of party seats can enhance the robustness of the veto power outcomes.
>
> In one of our task scenarios, namely the development subtopic with 4 parties, we randomly generated 30 different party seat allocations for each data point and conducted experimental analysis on the veto power objective. The results are shown as follows:
>
> **The VP score with 30 samples for each issue on the development topic.**
> | Model         | VP Score   |
> |---------------|------|
> | Qwen2.5-32B   | 0.53 $\pm$ 0.04 |
> | Llama-3.3-70B | 0.52 $\pm$ 0.04 |
> | Qwen2.5-72B   | 0.53 $\pm$ 0.03 |
> | GPT-4o        | 0.56 $\pm$ 0.04 |
> | Deepseek-V3.1 | 0.57 $\pm$ 0.04 |
> | Gemini-2.5    | 0.59 $\pm$ 0.03 |
>
> As shown in the table above, we present the mean and variance for different seat allocation outcomes. By examining the experimental results in the table, we find that they are entirely consistent with those reported in our Table 1: Qwen2.5-32B, Llama-3.3-70B, and Qwen2.5-72B exhibit similar performance, which is lower than that of the three commercial models. For these three commercial models, this capability, from strongest to weakest, is: Gemini-2.5, Deepseek-V3.1, and GPT-4o. This finding corroborates our first experimental conclusion, namely that increasing the number of seats sample held by a party will enhance, rather than diminish, the robustness of the veto power task.
>
> **Experiment 2**: We can conduct the analysis by examining the variances in the table above. It can be seen that all the variances are kept within a controllable range. This result demonstrates that although different choices for the number of party seats do indeed affect the outcomes of the veto power task to some extent, the impact is not significant and remains within a relatively controllable range.
>
> Combining the two experiments above, we arrive at the following two conclusions:
> 1. **Sampling different numbers of seats for each party can significantly enhance the robustness of the experimental results on veto power**;
> 2. **The choice of specific party seat allocations has only a minor impact on the final results.**
>
> We have already updated this part in the newly added Appendix F.2 in our revised manuscript.

---

> ### Author Response · Authors · 2025-11-20
>
> > **Q8. Could you provide a simple and intuitive baseline, such as greedy strategy?**
>
> **A8**: Thank you for your question. Following your suggestion, we have added two different baselines, namely **Random** and **Greedy**, allowing you to understand the performance level of current LLMs in a more intuitive way. The **Random** method randomly selects one party’s stances as the resolution, while the **Greedy** method selects the stances of the party with the largest number of seats as the resolution. We have conducted experiments across all settings, and the results are shown in the table below:
>
> **Performance of different Methods and LLMs on PoliCon.**
> | Model         | SM-2 | SM-4 | SM-6 | 2/3M-2 | 2/3M-4 | 2/3M-6 | VP-2 | VP-4 | VP-6 | Rawls-2 | Rawls-4 | Rawls-6 | Util-2 | Util-4 | Util-6 |
> |---------------|------|------|------|--------|--------|--------|------|------|------|---------|---------|---------|--------|--------|--------|
> | Random        | 0.56 | 0.53 | 0.56 | 0.29   | 0.20   | 0.14   | 0.36 | 0.35 | 0.38 | 2.59    | 2.01    | 1.77    | 5.04   | 4.78   | 4.80   |
> | Greedy        | 0.80 | 0.74 | 0.73 | 0.45   | 0.37   | 0.28   | 0.46 | 0.44 | 0.44 | 2.61    | 2.02    | 1.74    | 5.07   | 4.79   | 4.79   |
> | Qwen2.5-32B   | 0.74 | 0.80 | 0.87 | 0.34   | 0.39   | 0.40   | 0.47 | 0.55 | 0.62 | 4.02    | 3.50    | 3.19    | 6.01   | 6.27   | 6.38   |
> | Llama-3.3-70B | 0.72 | 0.78 | 0.86 | 0.37   | 0.45   | 0.48   | 0.46 | 0.55 | 0.63 | 3.98    | 3.42    | 3.11    | 6.08   | 6.40   | 6.56   |
> | Qwen2.5-72B   | 0.76 | 0.82 | 0.88 | 0.40   | 0.47   | 0.49   | 0.50 | 0.57 | 0.65 | 4.11    | 3.46    | 3.13    | 6.11   | 6.39   | 6.53   |
> | GPT-4o        | 0.83 | 0.87 | 0.92 | 0.51   | 0.57   | 0.63   | 0.54 | 0.62 | 0.69 | 4.50    | 3.80    | 3.42    | 6.40   | 6.62   | 6.80   |
> | Deepseek-V3.1 | 0.87 | 0.89 | 0.93 | 0.52   | 0.57   | 0.63   | 0.58 | 0.64 | 0.71 | 4.52    | 3.78    | 3.42    | 6.38   | 6.62   | 6.77   |
> | Gemini-2.5    | 0.88 | 0.90 | 0.90 | 0.53   | 0.57   | 0.58   | 0.61 | 0.66 | 0.70 | 4.60    | 3.91    | 3.51    | 6.39   | 6.56   | 6.68   |
>
> As shown in the table above, **Random** attains the lowest scores across all tasks. This indicates that current LLMs still possess some ability to seek political consensus, rather than generating text in a purely random manner, at least for the tasks we evaluate. As for **Greedy**, it outperforms the Random method on most tasks but is generally inferior to all LLM-generated consensus texts. However, on the SM and 2/3M tasks under the 2-party setting, it surpasses Qwen2.5-32B, Llama-3.3-70B, and Qwen2.5-72B, ranking second only to GPT-4o; on the VP task, its performance is comparable to that of Llama-3.3-70B. This suggests that most current open-source models still lag behind the Greedy baseline on these tasks. A possible reason is that these models attempt to satisfy both major and minor parties simultaneously, ultimately leading to outcomes that satisfy neither. Our experiments further demonstrate that these models still have substantial room for improvement on such tasks.
>
> In our revised manuscript, the complete experimental results have been added to Table 1, the descriptions of the two baselines have been added to lines 334–337, and the corresponding experimental analysis has been added to lines 451–457.
>
>
>
>
> **Reference:**
>
> [1] Scott D W. Multivariate density estimation: theory, practice, and visualization.

---

> > ### Comment · Reviewer_u21o · 2025-11-23
> >
> > I thank the authors for their efforts to address the questions and concerns I raised in my review in a detailed manner. I believe that the additional experiments and explanations provided by the authors considerably improve the quality of the manuscript. Therefore, I will increase my score accordingly.

---

> > > ### Author Response · Authors · 2025-11-24
> > >
> > > Thank you for your acknowledgment!

---

### Official Review · Reviewer_XoDH · 2025-10-28

**Soundness:** 3
**Presentation:** 4
**Contribution:** 3
**Rating:** 6
**Confidence:** 4

**Summary:**

This paper proposes PoliCon, a new benchmark to evaluate how large language models (LLMs) handle political consensus building. The authors collect real parliamentary data to create diverse decision-making scenarios and define several types of consensus goals. They design an LLM-based evaluation framework that measures whether a model’s generated resolutions can reach agreement among simulated parties. Experiments show that current LLMs perform well on simple majority decisions but struggle with more complex or fairness-oriented objectives. The results also reveal clear topic sensitivity and political bias, indicating that existing models still lack balanced reasoning in social and political contexts. Overall, the work introduces a valuable and well-structured framework for assessing LLMs’ ability to reason about group decisions and fairness.

**Strengths:**

1. The paper tackles a fresh and meaningful problem and it builds a solid and realistic benchmark using real parliamentary data, making the evaluation credible and grounded.

2. The evaluation setup is thoughtfully designed and connects well with social choice theory.

3. The experiments are thorough and provide clear insights into where current models perform well and where they fail.

**Weaknesses:**

1. The evaluation still depends on another LLM as a judge, which could introduce hidden bias.

2. The dataset only covers European Parliament data, so it might not generalize to other regions or political systems.

3. There’s no human validation to confirm that the evaluation results truly match real consensus reasoning.

4. Using LLMs in both dataset creation and evaluation could lead to subtle data leakage.

**Questions:**

1. How do you make sure the LLM-based judge is fair and not favoring certain model types?

2. Did you involve any human experts to check whether the model’s “consensus” decisions make real-world sense?

3. How could this benchmark be extended to other political or cultural settings beyond Europe?

4. What steps did you take to avoid overlap or leakage between LLM-generated data and evaluation content?

---

> ### Author Response · Authors · 2025-11-20
>
> Thank you, reviewer XoDH, for your constructive feedback. Here are our responses, organized to address each of your concerns one by one, specifically:
>
> > **Q1. How can we ensure that LLM-based evaluation is fair?**
>
> **A1**: Thank you for your question. We have incorporated a detailed human study to address your concern. Because our task involves both AI and political science, we recruit 20 human annotators. Half of them have AI-related backgrounds, and the other half are law students, allowing the team to cover a broader range of relevant domain knowledge.
>
> We randomly sample 100 consensus resolutions generated by LLMs for evaluation. To ensure annotation robustness, each data item is assigned to three different annotators, yielding a total of 300 annotation results. Consistent with the evaluator's scoring method, we use the expected values of consistency and feasibility as the final evaluation scores. We present the consistency results of our evaluator compared with the ground truth and human evaluations in the following table:
>
> **Consistency of our evaluator with ground truth and human evaluations.**
> |                | Ground truth | Human eval. |
> |:----------------:|:--------------:|:-------------:|
> | Mean error     | 1.36         | 1.61        |
> | σ              | 1.90         | 1.92        |
> | Within ±σ      | 72%          | 72%         |
>
> We report the mean error, the standard deviation $\sigma$, and the proportion of samples whose errors fall within $\pm \sigma$. It can be seen that the mean error between our evaluator and human annotations is only 1.61, and more than 72% of the data errors fall within $\pm 1.92$, which demonstrates that our evaluator is sufficient to generalize in evaluating LLM-generated consensus resolutions. Another interesting phenomenon is that the consistency between our evaluator and human annotators is highly aligned with the consistency between our evaluator and the real-world ground-truth voting results, indicating that our consistency-computation method based on real voting data is also effective.
>
> This part has been detailed in lines 371-416 of Section 4.1 and the newly added Appendix E.2 in our revised manuscript.
>
> > **Q2. This dataset only covers data from the European Parliament, so it may not be generalizable to other regions or political systems. How can this standard be extended to political or cultural contexts outside of Europe?**
>
> **A2**: Thank you for your question. **Regardless of the specific structure of a legislative body, the essence of political consensus-building remains the same, that is: reaching agreement among stakeholders under specific rules.** In general, this is done through majority voting, primarily using simple majority or two-thirds majority rules. This can be observed in the regulations of many political decision-making bodies outside of the European Parliament. For example, Article 18 of the United Nations Charter [1] stipulates that important resolutions in the UN General Assembly require a two-thirds majority vote, while other matters are decided by a simple majority of the members present and voting. The U.S. House of Representatives [2] and the Canadian House of Commons [3] also adopt simple majority voting to pass resolutions. In addition, the Japanese House of Councillors [4] and the UK House of Commons [5] explicitly provide for majority voting as the method to determine whether consensus is achieved.
>
> Moreover, **PoliCon itself is designed to generalize beyond the European Parliament**. It supports a flexible number of stakeholders and combines veto, utilitarian and fairness-oriented objectives, inspired by systems like the UN Security Council and social choice theory. This design allows users to simulate diverse consensus-seeking scenarios, e.g., two-party dynamics similar to the U.S. Democratic and Republican parties.
>
> Certainly, we greatly appreciate your suggestion to incorporate different cultural backgrounds, as this can provide new insights for our future research.

---

> ### Author Response · Authors · 2025-11-20
>
> > **Q3. Human verification is needed to check whether the consensus resolutions align with real-world logic.**
>
> **A3**: Thank you for your question. However, as stated in lines 246–248 of our paper, an important metric in our evaluation is whether the consensus decisions generated by the model align with real-world principles. If a generated resolution violates real-world logic, it receives a lower score; conversely, if it is logically sound, it receives a higher score. Therefore, in response to your question, what truly matters is whether our evaluator can reliably assess the real-world feasibility of the model-generated consensus.
>
> To verify this, we conducted a human study. As in our response to your Q1, we asked the same 20 annotators to rate whether each model-generated resolution was consistent with real-world logic, i.e. its feasibility. Based on 300 annotated samples, the mean error was only 1.61, and more than 70.7% of the data points fell within $\pm \sigma = 2.12$, demonstrating that our evaluator is sufficiently capable of determining whether the model-generated resolutions adhere to real-world logic.
>
> > **Q4. Using LLMs in dataset creation and evaluation may lead to subtle data leakage. What methods have you taken to prevent leakage between LLM-generated data and evaluation content?**
>
> **A4**: Thank you for your question. **The main source of potential leakage is whether the European Parliament data appeared in the LLM’s pretraining, not our use of LLMs for cleaning or evaluation**. To mitigate this, we randomize each party’s seats, allow flexible numbers of parties, and introduce diverse political objectives (veto power, Rawlsian, Utilitarian). This approach not only ensures task configurations differ from real-world data while retaining realism (as detailed in lines 287–315), but also exposes LLM biases: if an LLM favors certain parties, it may fail to reach the required consensus, which is reflected in results (lines 2335–2339). Our experiments show state-of-the-art LLMs struggle with complex tasks like 2/3M, suggesting the impact of data leakage might not be significant.
>
> Additionally, it should be clarified that we did not use LLMs during dataset creation, rather, LLMs were employed in the data cleaning process. This process does not cause data leakage, but it may introduce the risk of LLM bias. To assess this, we conducted a human evaluation with 10 annotators on 100 samples. Results show over 95.8% of resolutions and 75% of stances scored over 3 out of 5, indicating that LLM-based cleaning maintains acceptable reliability.
>
> The human study has been detailed in our revised manuscript in lines 1785–1794 of Appendix D, as well as in the newly added Appendix E.2.
>
> **Reference:**
>
> [1] United Nations. Charter of the United Nations. (https://www.un.org/en/about-us/un-charter/full-text)
>
> [2] U.S. House of Representatives. The legislative process. (https://www.house.gov/the-house-explained/the-legislative-process)
>
> [3] House of Commons of Canada. "Our Procedure — Debate, Voting and Decorum". (https://www.ourcommons.ca/procedure/our-procedure/DebateandVoting/c_g_debatevoting-e.html)
>
> [4] House of Councillors, The National Diet of Japan. Organization. (https://www.sangiin.go.jp/eng/guide/organ/index1.htm)
>
> [5] House of Commons Information Office. Divisions. (https://www.parliament.uk/globalassets/documents/commons-information-office/Divisions.pdf)

---

> > ### Comment · Reviewer_XoDH · 2025-11-26
> >
> > Thank you for the detailed response. My concerns have been well addressed, and I will maintain my positive score.

---

> > > ### Author Response · Authors · 2025-11-27
> > >
> > > Thank you for your response. We are genuinely grateful for your thoughtful feedback, which further improves the clarity of our paper.

---

### Official Review · Reviewer_cGY7 · 2025-10-30

**Soundness:** 4
**Presentation:** 3
**Contribution:** 3
**Rating:** 8
**Confidence:** 3

**Summary:**

This paper introduces PoliCon, a novel benchmark for evaluating LLMs' ability to achieve political consensus objectives. Built from 2,225 European Parliament deliberation records (2009-2022), PoliCon tests LLMs across diverse collective decision-making scenarios incorporating political issues, goals, participating parties, and power structures. The benchmark defines five distinct political consensus objectives (Simple Majority, Two-thirds Majority, Veto Power, Rawlsianism, and Utilitarianism) and employs an evaluation framework based on social choice theory that simulates real voting outcomes. The authors evaluate six state-of-the-art LLMs, revealing significant performance gaps in complex consensus scenarios and uncovering inherent partisan biases.

**Strengths:**

1. First benchmark specifically designed to evaluate LLMs' political consensus-building capabilities across diverse objectives
2. 2,225 high-quality parliamentary records with extensive cleaning and processing, integrating multiple sources
3. Diverse task settings: 15 different configurations combining party numbers, voting mechanisms, and political goals, creating 28,620 distinct scenarios

**Weaknesses:**

1. Using GPT-4o-mini as evaluator could introduce circular biases when testing other LLMs

**Questions:**

Could the authors clarify whether the GPT-4o-mini evaluator was temperature-controlled (e.g., deterministic setting) during scoring, and whether variance across seeds was analyzed?

---

> ### Author Response · Authors · 2025-11-20
>
> Thank you, reviewer cGY7, for your constructive feedback and acknowledgement. Here are our responses, organized to address each of your concerns one by one, specifically:
>
> > **Q1. Will using GPT-4o-mini as an evaluator introduce bias?**
>
> **A1**: Thank you for your question. We have incorporated a detailed human study to address your concern. Because our task involves both AI and political science, we recruit 20 human annotators. Half of them have AI-related backgrounds, and the other half are law students, allowing the team to cover a broader range of relevant domain knowledge.
>
> We randomly sample 100 consensus resolutions generated by LLMs for evaluation. To ensure annotation robustness, each data item is assigned to three different annotators, yielding a total of 300 annotation results. Consistent with the evaluator's scoring method, we use the expected values of consistency and feasibility as the final evaluation scores. We present the consistency results of our evaluator compared with the ground truth and human evaluations in the following table:
>
> **Consistency of our evaluator with ground truth and human evaluations.**
> |                | Ground truth | Human eval. |
> |:----------------:|:--------------:|:-------------:|
> | Mean error     | 1.36         | 1.61        |
> | σ              | 1.90         | 1.92        |
> | Within ±σ      | 72%          | 72%         |
>
> We report the mean error, the standard deviation $\sigma$, and the proportion of samples whose errors fall within $\pm \sigma$. It can be seen that the mean error between our evaluator and human annotations is only 1.61, and more than 72% of the data errors fall within $\pm 1.92$, which demonstrates that our evaluator is sufficient to generalize in evaluating LLM-generated consensus resolutions. Another interesting phenomenon is that the consistency between our evaluator and human annotators is highly aligned with the consistency between our evaluator and the real-world ground-truth voting results, indicating that our consistency-computation method based on real voting data is also effective.
>
> This part has been detailed in lines 371-416 of Section 4.1 and the newly added Appendix E.2 in our revised manuscript.
>
> > **Q2. Did the evaluator implement temperature control during the scoring process, and did the authors analyze the differences between seeds?**
>
> **A2**: Thank you for your question. To ensure the determinism of the scoring results, we set the temperature of the evaluators to 0 during the scoring process. In our original setup, we had fixed a single random seed, 42. To address your concern, we have now used three different random seeds. Below, we show the evaluation results for one of our task scenarios, namely, 2/3M, "development" subtopic, and 4 parties. We present the mean and variance of the results across the different seeds, which are shown in the table below:
>
> **Results of more random seeds**
> | Model         |    Avg. & Std.  |
> |:--------------|:------------|
> | Qwen2.5-32B   |  0.47 $\pm$ 0.05 |
> | Llama-3.3-70B |  0.49 $\pm$ 0.02 |
> | Qwen2.5-72B   |  0.68 $\pm$ 0.00 |
> | GPT-4o        |  0.57 $\pm$ 0.02 |
> | Deepseek-V3.1 |  0.61 $\pm$ 0.00 |
> | Gemini-2.5    |  0.57 $\pm$ 0.02 |
>
> As the table shows, despite minor differences, the impact of different random seeds on the evaluation results is within a controllable range.

---

> > ### Comment · Reviewer_cGY7 · 2025-11-26
> >
> > I thank the authors for their efforts to address the raised points. My concerns have been resolved, and I keep my score.

---

> > > ### Author Response · Authors · 2025-11-26
> > >
> > > Thank you for your response. We are genuinely grateful for your thoughtful feedback, which further improves the clarity of our paper.

---

### Official Review · Reviewer_CDif · 2025-11-05

**Soundness:** 3
**Presentation:** 3
**Contribution:** 2
**Rating:** 4
**Confidence:** 5

**Summary:**

This paper evaluates LLMs' capabilities to generate consensus statements on political issues. The authors created a novel benchmark constructed from 2,225 high-quality deliberation records of the European Parliament over 13 years. With an automatic evaluation pipeline, this paper demonstrates that LLMs have varied capabilities on generating consensus statements and most of the models remain undersatisfied with complex tasks.

**Strengths:**

1. This paper tackles an important topic
2. The results are presented clearly

**Weaknesses:**

1. One of the major weaknesses of the paper is the reliability of the LLM-as-judge pipeline. LLM-as-judge has long been criticized for its generalizability, which applies to this paper as well. If I understand it correctly, the LLM-as-judge evaluation heavily relies on existing statements and voting data. However, it is really unclear whether this pipeline could create generalizable results for new statements. I believe this is the major issue of this paper. I'm happy to discuss with the authors about this.

2. A second weakness is regarding the motivation and practical value of this work. In parliament deliberations, it is the deliberation process that leads to the final voting results instead of the statement itself. Therefore, this task setting may not reflect real-world settings where the consensus statements are actually needed.

**Questions:**

please see weakness section

---

> ### Author Response · Authors · 2025-11-20
>
> Thank you, reviewer CDif, for your constructive feedback. Here are our responses, organized to address each of your concerns one by one, specifically:
>
> > **Q1. How reliable is the process of using LLMs as evaluators, and can this process be generalizable to new statements?**
>
> **A1**: Thank you for your question. Firstly, we need to clarify that although the experiments in Section 4.1 of our paper do use existing statements and voting data, for the LLM, these are entirely new statements. Moreover, we conducted 41,800 validation experiments, which is a very substantial amount of data. Therefore, the concern you raised about whether the results can generalize to newly generated statements does not actually arise.
>
> Certainly, to further address your concerns, we have incorporated a detailed human study. Because our task involves both AI and political science, we recruit 20 human annotators. Half of them have AI-related backgrounds, and the other half are law students, allowing the team to cover a broader range of relevant domain knowledge.
>
> We randomly sample 100 consensus resolutions generated by LLMs for evaluation. To ensure annotation robustness, each data item is assigned to three different annotators, yielding a total of 300 annotation results. Consistent with the evaluator's scoring method, we use the expected values of consistency and feasibility as the final evaluation scores. We present the consistency results of our evaluator compared with the ground truth and human evaluations in the following table:
>
> **Consistency of our evaluator with ground truth and human evaluations.**
> |                | Ground truth | Human eval. |
> |:----------------:|:--------------:|:-------------:|
> | Mean error     | 1.36         | 1.61        |
> | σ              | 1.90         | 1.92        |
> | Within ±σ      | 72%          | 72%         |
>
> We report the mean error, the standard deviation $\sigma$, and the proportion of samples whose errors fall within $\pm \sigma$. It can be seen that the mean error between our evaluator and human annotations is only 1.61, and more than 72\% of the data errors fall within $\pm$1.92, which demonstrates that our evaluator is sufficient to generalize in evaluating LLM-generated consensus resolutions. Another interesting phenomenon is that the consistency between our evaluator and human annotators is highly aligned with the consistency between our evaluator and the real-world ground-truth voting results, indicating that our consistency-computation method based on real voting data is also effective.
>
> This part has been detailed in lines 371-416 of Section 4.1 and the newly added Appendix E.2 in our revised manuscript.

---

> ### Author Response · Authors · 2025-11-20
>
> > **Q2. In a parliament, the final voting outcome is determined by the deliberation process rather than by the resolution itself. Therefore, might this task setup fail to reflect real-world scenarios in which consensus statements are genuinely required?**
>
> **A2**: Thank you for your question. First, I would like to clarify that **in real-world parliaments, the outcome of a vote is determined solely by the text of the resolution itself**. This can be substantiated by the rules of various political decision-making bodies in the real world. For example, Rule 136 of the Rules of Procedure of the European Parliament [1], which states "… If Parliament decides to wind up a debate with a resolution, a committee, a political group or Members reaching at least the low threshold may table a motion for a resolution …", shows that the vote on the resolution text takes place only after sufficient deliberation. Rule 189 [2], which provides for "… voting on the motion for a resolution as a whole (final vote) …", indicates that the final step of the deliberative process is a vote on the complete text of the resolution. Moreover, this is not unique to the European Parliament: the Standing Orders of the House of Commons of Canada [3], the New Zealand Parliament [4], and the U.S. House of Representatives [5] all follow the same practice. Therefore, the task design of PoliCon can fully reflect the requirements of real-world scenarios in which consensus statements are genuinely needed.
>
> Regarding your point that the deliberation process may influence the voting outcome, I assume what you mean is that the process of generating a resolution requires the involvement of deliberation. This is correct. In reality, establishing a final resolution indeed requires multiple rounds of negotiation and revision. However, **the task we have set does not target the process of generating the resolution, but rather the LLM's ability to reach the consensus outcome**. Since producing a consensus resolution text during deliberation happens to be the least efficient part of how humans reach consensus, this difference becomes a key entry point for studying how AI can improve real-world political consensus-building and further enhance social governance. This approach not only allows us to orient AI research in this area toward improving the efficiency of social governance, but also gives the design greater generality: it can support high-level consensus formation, such as in parliaments, while also serving everyday community-level consensus-building needs, without being constrained by the specific form in which consensus is generated.
>
>
> **Reference:**
>
> [1] European Parliament. Rules of Procedure of the European Parliament (Rule 136). (https://www.europarl.europa.eu/doceo/document/RULES-10-2025-07-07-RULE-136_EN.html)
>
> [2] European Parliament. Rules of Procedure of the European Parliament (Rule 189). (https://www.europarl.europa.eu/doceo/document/RULES-10-2025-07-07-RULE-189_EN.html)
>
> [3] House of Commons of Canada. "Our Procedure: Debate, Voting and Decorum". (https://www.ourcommons.ca/procedure/our-procedure/DebateandVoting/c_g_debatevoting-e.html)
>
> [4] New Zealand Parliament. Chapter 18: Motions and Amendments. (https://www3.parliament.nz/en/visit-and-learn/how-parliament-works/parliamentary-practice-in-new-zealand-2023-by-chapter/chapter-18-motions-and-amendments)
>
> [5] Rules of the House of Representatives. Rule XIX: Motions Following the Amendment Stage. (https://clerk.house.gov/legislative/house-rules.pdf)

---

> ### Author Response · Authors · 2025-11-27
>
> Dear Reviewer CDif,
>
> Thank you for your insightful feedback and valuable comments on our paper. We have carefully addressed the concerns you raised, as outlined in the revised version. Could you kindly confirm if the revisions adequately address the issues? If there are any remaining points that require further discussion or clarification, we would be more than willing to engage and improve the work accordingly.
>
> Additionally, if you find the changes satisfactory, we would appreciate your consideration for adjusting the score to reflect the improvements.
>
> Looking forward to your feedback.

---

### Author Response · Authors · 2025-12-02
**Summary of the revised manuscript**

Dear Reviewers and Area Chair,

We thank all the reviewers for their valuable comments and suggestions on our paper. We have carefully revised the manuscript and conducted additional experiments in response to their feedback, with the revised parts highlighted in blue. For your convenience, we provide a summary of the modifications and updates made below:

1. **Human Verification** (@Reviewer CDif, cGY7, XoDH, u21o): We update Section 4.1 to include the settings and results of our human study. We also add Appendix E.2 to provide additional details on the experimental procedure.
2. **Detailed Judge Calculation** (@Reviewer u21o): We add a description of this process in line 1627 of Appendix C.3.
3. **Formula used in Figure 4** (@Reviewer u21o): We provide a detailed description of the formula for calculating and plotting the error curve in Figure 4 in line 362 and in the additionally added Appendix E.1.
4. **Specific Data Filtering Process** (@Reviewer u21o): We add the data filtering procedure after the data cleaning process with DeepSeek-R1 in Appendix A.6.
5. **Clarification of Veto Power's Definition** (@Reviewer u21o): We modify line 303 to correct our original wording, which might have been ambiguous regarding the definition of the veto power task setting.
6. **Effect of Party Seats** (@Reviewer u21o): We add two experiments in the newly added Appendix F.2, which explain in detail how the choice of party seats affects the robustness of the veto power task results.
7. **Additional Baselies** (@Reviewer u21o): We add two baselines, random and greedy; their complete experimental results are added to Table 1, the descriptions of the two baselines are added to lines 334–337, and the corresponding experimental analysis is added to lines 451–457.

We believe these updates adequately address all the reviewers' concerns and improve the quality of our manuscript. We sincerely appreciate your time and consideration of our revised manuscript.

Best,

The Authors

---

### Author Response · Authors · 2025-12-02
**Notes to AC: Summary of the Rebuttal**

Dear Area Chair,

We sincerely appreciate your stepping in to oversee our submission under these unprecedented circumstances. We understand the significant additional effort required to reevaluate the manuscript and discussion history, and we are grateful for your time and dedication to the scientific integrity of the conference. To assist in your assessment, we will briefly summarize our rebuttal discussion and explain how we have addressed all the reviewers' concerns.

We appreciate the reviewers' valuable feedback and the efforts they have devoted. **Before this unexpected incident**, reviewer u21o believed that **our additional experiments and explanations greatly improved the quality of our manuscript and had already raised the score from 6 to 8**. Reviewers cGY7 and XoDH **both stated that their concerns had been fully resolved and were willing to maintain their positive scores**. However, we are concerned about the lack of engagement from reviewer CDif during the discussion phase, which **risks leaving our work insufficiently evaluated, despite the fact that we have addressed all the questions raised**. We hope this does not compromise a fair assessment of our submission. Below, we summarize how we addressed each of the reviewer CDif's two concerns:

1. **Evaluation Reliability**: We recruit 20 annotators (10 with AI backgrounds and 10 law students) to rate 100 LLM-generated consensus resolutions, each annotated by three people, and compare our evaluator's scores with human judgments. We report the mean error, the standard deviation $\sigma$, and the proportion of samples whose errors fall within $\pm\sigma$. It can be seen that **the mean error between our evaluator and human annotations is only 1.61, and more than 72% of the data errors fall within $\pm$1.92, which demonstrates that our evaluator is sufficient to evaluate LLM-generated consensus resolutions**.

2. **Consistency with Real-World Scenarios**: We show that, **in major legislatures, the final decision is made by voting on the complete text of a resolution, which is consistent with our task setup**, as codified in the Rules of Procedure of the European Parliament (Rules 136 and 189) and the standing orders of the Canadian, New Zealand, and U.S. Houses. Building on this, we emphasize that PoliCon is designed to evaluate the LLM's ability to reach the consensus outcome given a final resolution text, rather than to simulate the entire deliberation process, which **mirrors the decisive step in real political decision-making while remaining flexible enough to cover both high-level parliamentary settings and everyday community consensus-finding**. These clarifications demonstrate that **PoliCon both faithfully captures how consensus resolutions operate in real-world institutions and targets precisely the stage at which LLMs can most effectively support political and social governance**.

Additionally, it is worth noting that **the two questions raised by reviewer CDif are also brought up by other reviewers, all of whom indicated that they have been addressed**. In particular, question 1 is mentioned by all four reviewers, and question 2 is almost identical to question 2 raised by reviewer XoDH.

We hope this summary enables a fair assessment, mitigating the impact of reviewer CDif's lack of engagement and the disruption caused by the unexpected incident on the rebuttal discussion.

Thank you for your time and effort.

Best regards,

Submission 4014 Authors

---

### Meta-Review · Area_Chair_sbNx · 2026-01-06

**Summary:**

Reviewers agreed that the problem tackled in this paper is highly significant, and appreciated the novelty of the benchmark that is based on real-world data. While multiple reviewers raised concerns on reliability of the LLM-as-judge pipeline and the practical value of this work, the rebuttal appears to be satisfactory.

**Reviewer Concerns:**

The main concerns are from CDif and some other reviewers also raised the same questions. (1) reliability of the LLM-as-judge pipeline and (2) the practical value of this work. Reviewer CDif was open for discussions but unfortunately didn't get a chance to do so before the chaos. The rebuttal appears to be rigorous but I am not able to judge its correctness. Personally I don't think these two questions can be fully addressed, but I'd intend to accept the paper due to its novelty and relative complete execution of ideas.

**Reviewer Scores:**

CDif might raise score

---

### Decision · Program_Chairs · 2026-01-26

Accept (Poster)